# On the equilibrium limit of liquid stability in pressurized aqueous systems

Arian Zarriz[1], Baptiste Journaux [2] ✉ & Matthew J. Powell-Palm [1,3,4] ✉

Phase stability, and the limits thereof, are a central concern of materials thermodynamics. However, the temperature limits of equilibrium liquid stability in chemical systems have only been widely characterized under constant (typically atmospheric) pressure conditions, whereunder these limits are represented by the eutectic. At higher pressures, the eutectic will shift in both temperature and chemical composition, opening a wide thermodynamic parameter space over which the absolute limit of liquid stability, i.e., the limit under arbitrary values of the thermodynamic forces at play (here pressure and concentration), might exist. In this work, we use isochoric freezing and melting to measure this absolute limit for the first time in several binary aqueous brines, and nodding to the etymology of "eutectic", we name it the "cenotectic" (from Greek "κοινός-τῆξῐς", meaning "universal-melt"). We discuss the implications of our findings on ocean worlds within our solar system and cold ocean exoplanets; estimate thermodynamic limits on ice crust thickness and final ocean depth (of the cenotectic or "endgame" ocean) using measured cenotectic pressures; and finally provide a generalized thermodynamic perspective on (and definition for) this fundamental thermodynamic invariant point.

Multiphase liquid-solid equilibrium ranks amongst the most fundamental concepts in thermodynamics and physical chemistry, providing the foundation of modern phase diagrams, grounding thermodynamic analysis, and driving countless industrial processes. In multi-component systems, at constant pressure, the lowest temperature at which a liquid may remain stable at equilibrium is defined as the eutectic (from Greek "εὐ-τῆξῐς", "easy-melt"), an invariant point in temperature-concentration (T-x) space at which the liquid will transition entirely into a mixture of solid phases. Eutectics are of crucial importance to a wide range of applications, from metallurgy to igneous rock formation, phase change thermal energy storage to cryopreservation, drug discovery to planetary science, etc. The canonical definition of the eutectic is often considered at a fixed pressure (typically 0.1 MPa / atmospheric pressure); however, increased states of compression may substantially affect the melting curves of solids, resulting in a change of eutectic T-x coordinates. When pressure is varied, the eutectic becomes an univariant curve in P-T-x space, the trajectory of which depends upon the geometry of the liquidus curves of the contributing solid phases.

For most materials, solid phases are denser than the liquid phase, meaning these materials possess positive Clapeyron melting slopes (dP/dT > 0). For systems containing such solids, univariant eutectic curve (also called cotectic) temperatures will increase with increasing pressures.

Select compounds in natural science and engineering, however, such as water, silicon, or gallium, have negative Clapeyron melting slopes at 0.1 MPa (i.e., select solid phases of these compounds are less dense than the liquid). In many binary systems containing these compounds, initial compression from 0.1 MPa leads to pressure-induced melting point depression, resulting in the univariant eutectic curve decreasing in temperature with increasing pressure[1]. However, for all of these systems, continued compression of the liquid will

[1]J. Mike Walker '66 Department of Mechanical Engineering, Texas A&M University, College Station, TX, USA. [2]Department of Department of Earth and Space Sciences, University of Washington, Seattle, WA, USA. [3]Department of Materials Science & Engineering, Texas A&M University, College Station, TX, USA. [4]Department of Biomedical Engineering, Texas A&M University, College Station, TX, USA. ✉e-mail: bjournau@uw.edu; powellpalm@tamu.edu

eventually result in the formation of a denser solid phase (e.g., ice III, Si-II, Ga-II), thereby reversing the Clapeyron slope. In this context, the negative-Clapeyron eutectic curve for the lower pressure range will intersect a positive-Clapeyron higher-pressure melting line at an invariant point in P-T-x space.

For aqueous solutions in which the eutectic in equilibrium with ice-Ih is the lowest-temperature eutectic in the phase diagram (including most solutions of salts, sugars, and other solutes that are solid at room temperature), this invariant point represents the lowest possible temperature at which a given aqueous liquid phase may remain stable at equilibrium under any P-T-x conditions and therefore represents a fundamental property of the system. While select previous works have inadvertently observed this fundamental invariant thermodynamic point in the study of high-pressure eutectics[2–4], to our knowledge, no previous study has identified or measured the invariant point itself, nor defined it as an entity distinct from other univariant transitions.

Based on the etymology of eutectic (from Greek "εὐ-τῆξις", "easy-melt"), we propose the name cenotectic (from Greek "κοινός-τῆξις", "universal-melt") for this invariant point, and we illustrate it in Fig. 1 as point κ (the Greek letter kappa), which presents a semi-quantitative P-T-x diagram for a binary system of water and a generic salt-like solute. Note that in most aqueous systems, given the geometries of the ices-liquid liquidus surfaces, this point is typically located on the ice Ih-ice III-liquid or ice Ih-ice II-liquid equilibrium boundary.

In this work, we describe first-of-their-kind experimental determinations of the precise P-T coordinates of the cenotectic points of major binary aqueous systems relevant to planetary sciences, medical sciences, and engineering. These results are acquired via an isochoric freezing and melting approach previously established by our group[1] for the interrogation of univariant phase configurations, which we herein extend to the absolute temperature limit of aqueous liquid stability and furthermore use to identify several potentially new high-pressure hydrate phases. We discuss the implications of this data on our evolving understanding of low-temperature aqueous

thermodynamics, detail implications in planetary science for icy ocean worlds, and provide a suggested roadmap toward rapid illumination of the yet-unexplored low-temperature high-pressure parameter space for aqueous solutions. In closing, we provide a generalized and complete definition of the cenotectic, applicable to arbitrary chemical systems under the influence of any arbitrary modes of thermodynamic work.

## Results

Here, we use isochoric freezing and melting[1] to measure the pressure-temperature evolution of the eutectic for eight aqueous binary solutions ($Na_2CO_3$, KCl, $MgSO_4$, $Na_2SO_4$, Urea, NaCl, $MgCl_2$, and $NaHCO_3$) in the temperature range 203–273.15 K and the pressure range 250 to 0.1 MPa, identifying the P-T coordinates of the cenotectic point in five of them and pointing at new possible high-pressure hydrate phases in the remaining three (NaCl, MgCl2, $NaHCO_3$). The eutectic P-T curves for all eight solutions studied are shown in Fig. 2.

In brief, the experimental methodology consists of confining each ~ 5.33 mL eutectic solution sample in a custom metallic isochoric chamber free of air and fitted with a high-accuracy pressure transducer; submerging the chamber in a calibration-grade programmable circulating bath; cooling the chamber continuously to a temperature at least 30 K beneath the 0.1 MPa eutectic temperature; allowing all crystallization processes to reach steady-state (after nucleation of ice III or ice II), as indicated by constancy of the pressure signal; then warming the chamber and melting its contents in 0.5 K increments, recording the steady pressure at each temperature. Isochoric conditions allow the pressure in the system to vary freely in response to the changing volumes of emerging or receding phases and couple the temperature and pressure according to Gibbs' phase rule, which dictates that a univariant phase configuration (i.e., the three-phase eutectic in a binary solution) has only one intensive degree of freedom.

Additional details on the isochoric freezing and melting process are provided in the Methods and in Supplementary Note 1, with chamber schematics provided in Supplementary Fig. 1. Example time-series

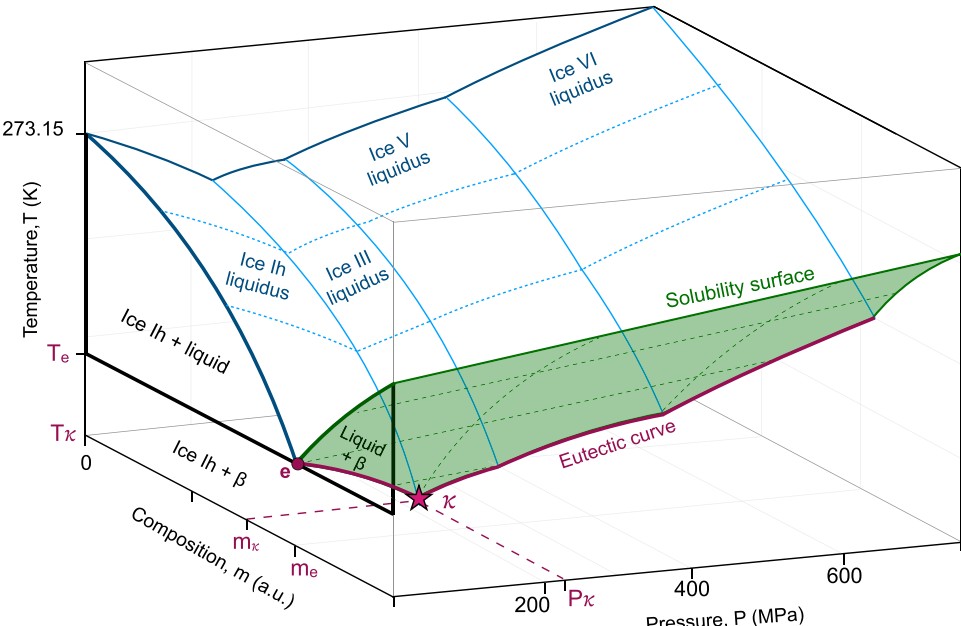

**Fig. 1 | Conceptual pressure-temperature-concentration phase diagram for a binary solution of water and a generic salt-like solute with only one solute-bearing solid phase.** The shaded green region shows the solubility surface of the solute. The eutectic at ambient pressure is marked by the magenta circle (e), the pressure-dependent eutectic curve is marked by the thick magenta line, and the cenotectic point (κ) is marked as a magenta star. $m_e$ marks the brine concentration at the 0.1 MPa eutectic, and $m_κ$ marks the brine concentration at the cenotectic point. $P_κ$, $T_κ$ mark the pressure and temperature of the cenotectic point, which we measure in this work. Isochoric freezing/melting of brine samples of concentration $m_e$ generally follows the pressure-temperature trajectory marked by the magenta eutectic curve between points (e) and (κ), and the cenotectic point can be identified by the change in the direction of the P-T slope of this curve.

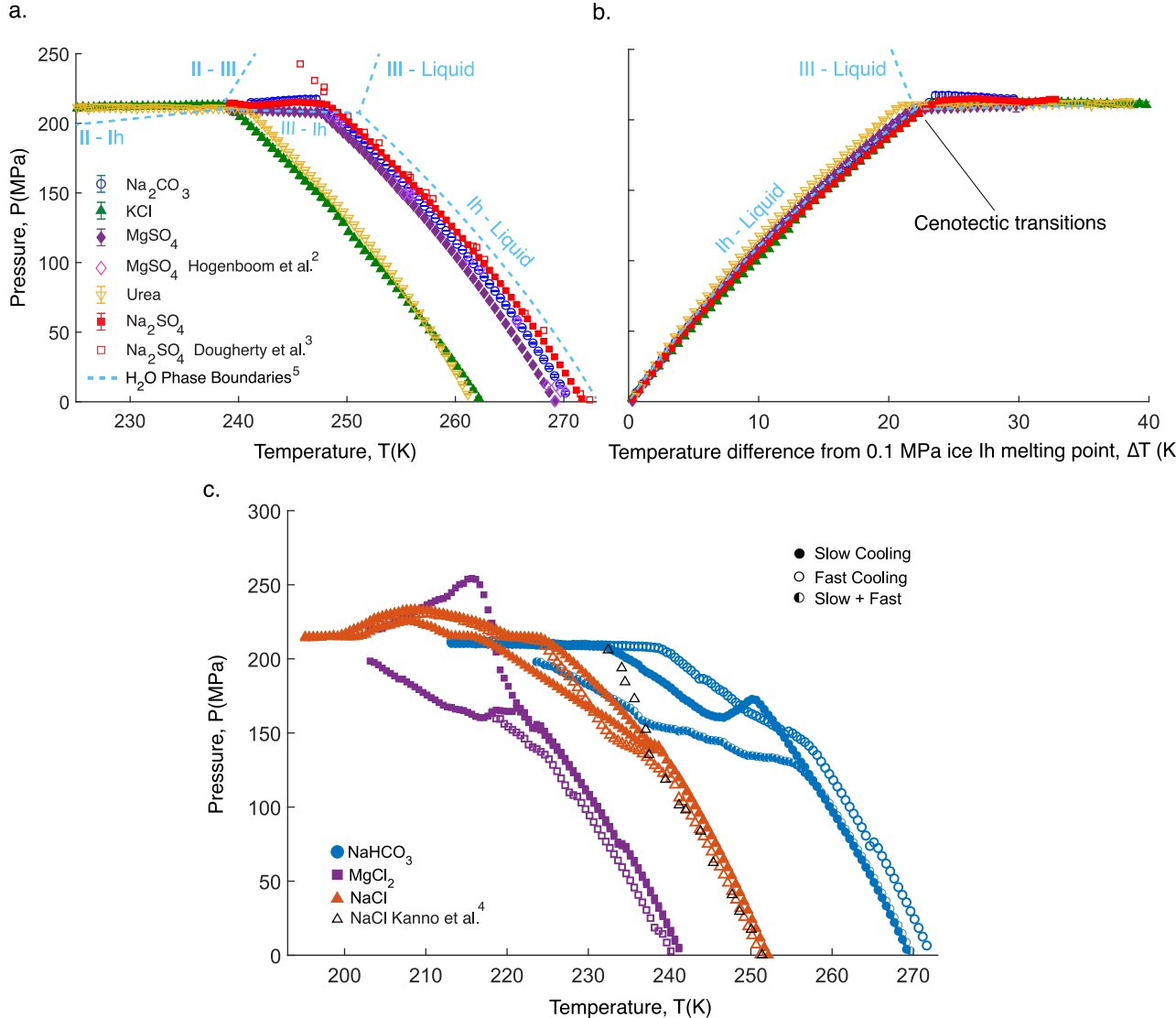

**Fig. 2 | Binary solution eutectic pressure-temperature (P-T) profiles. a** P-T profile for the five binary eutectic solutions tested that do not produce new salt-bearing solid phases at high pressure. The cenotectic point for each solution, or the limit of liquid stability, occurs at the slope discontinuity in each curve, at which point the remaining liquid solidifies into a solid solution of ice Ih, ice III, or ice II, and the salt-bearing solid phase. The phase boundaries of the pure water phase diagram (computed using the SeaFreeze thermodynamic framework[5]) are superimposed atop the data for context. **b** the same pressure profiles plotted against the temperature delta from the melting point of ice Ih, which for each of the solutions is the atmospheric pressure eutectic point, and for pure water is 273.15 K. In (**a**) and (**b**), markers present the mean of $n = 3$ replicate trials, and error bars show standard deviation (which is on the order of ~1 MPa across trials). **c** P-T profiles for three binary eutectic solutions that do produce new (potentially metastable) salt-bearing solid phases at high-pressure. With decreasing temperature, discontinuities in the slope of the P-T eutectic curve represent phase transitions involving the salt. Each curve represents a single trial, produced via either a slow cooling, fast cooling, or combined slow and fast cooling routine (details in "Methods"). Further detail on these trials is available in Supplementary Notes 2 and 3, and additional data recorded for the solutions producing intermediate phases is provided in Supplementary Fig. 3–5. Source data are provided in the Source Data file.

pressure-temperature data of the entire isochoric freezing and melting process are provided in Supplementary Note 2. Detailed uncertainty analysis is provided in Supplementary Note 4. Additional discussion of the thermodynamic principles, merits, and drawbacks of the technique is provided in Chang et al.[1].

## Cenotectic measurements

After total freezing of the solution, most experiments are observed to stabilize around 210 MPa (see, for example, Supplementary Fig. S2), coincident with the solid-solid phase transitions of ice Ih-ice III and ice Ih-ice II in the studied temperature range[5,6]. Upon warming, the pressure first remains approximately constant for some period, again consistent with the quasi-isobaric nature of the ice Ih-ice III/II phase boundary, and we then observe two varieties of isochoric pressure-

temperature curves, monotonic (Na$_2$CO$_3$, KCl, MgSO$_4$, Na$_2$SO$_4$, and Urea), and non-monotonic (NaCl, MgCl$_2$, and NaHCO$_3$).

For monotonic curves, shown in Fig. 2a, b, as temperature increases during warming, we first observe a clear discontinuity in the slope of the P-T curve upon the emergence of the liquid phase, and then a monotonic decrease in pressure as the liquid phase fraction increases along the eutectic curve. This first discontinuity is identified as the cenotectic point, the lowest temperature at which the liquid phase remains stable, and occurs at pressures consistent with the ice Ih-ice III or ice Ih-ice II transitions.

In order to precisely identify the cenotectic P-T coordinates, we fit polynomials to P-T eutectic curves colder and warmer than the apparent cenotectic point (the colder curve including ice Ih, ice II/III, and the salt-bearing solid phase; and the warmer curve including ice Ih,

the salt-bearing solid phase, and the liquid) and calculated their intersection, at which all four phases may coexist. Average cenotectic pressures and temperatures across $n = 3$ trials per solution are listed in Table 1, alongside the initial concentration of the solution used, the eutectic temperature measured at atmospheric pressure, and the temperature delta between the eutectic and the cenotectic. Propagated uncertainties accounting for both experimental error and fit uncertainty are included in parentheticals (details in Supplementary Note 4).

In Fig. 2a, we also compare our eutectic curve P-T results for $MgSO_4$ and $Na_2SO_4$ to those of Hogenboom and colleagues[2,3], which agree well and represent the only other data we were able to identify on high-pressure, low-temperature eutectic equilibria for the systems studied here. Furthermore, although Hogenboom and co. did not measure the cenotectic points of these two solutions, we may estimate them by quadratically fitting the data clusters along the ice Ih eutectic and ice III eutectic curves and calculating the intersections of these fits. The cenotectic values calculated from the data of Hogenboom and co. ($MgSO_4$: 211.74 MPa, 248.50 K; $Na_2SO_4$: 208.34 MPa, 249.95 K) stand in strong agreement with the measurements featured in Table 1.

These literature data also highlight the distinct enhancement in resolution afforded by the isochoric freezing and melting approach, which enables continuous generation and measurement of pressure in response to continuous variation of the temperature, as compared to isobaric techniques performed at discrete isothermal-isobaric pressure-temperature steps. The results herein furthermore confirm the suspicions of Hogenboom et al. that their measurements of water-$Na_2SO_4$ eutectic coordinates did not reflect the equilibrium state (i.e., entered a metastable configuration) at pressures above 200 MPa.

It should be noted that while we do not directly measure the evolution of the salt concentration in the aqueous liquid phase, based on our previous study of high-pressure binary eutectics[1], we expect that it will continuously decrease with increasing pressure, beginning at approximately the eutectic concentration ($m_e$ in Fig. 1; exact value listed in Table 1) and ending at the cenotectic concentration ($m_k$ in Fig. 1). Using the cenotectic P-T coordinates shown in Table 1, future work should aim to characterize the cenotectic salt concentration in the liquid directly.

For NaCl, $MgCl_2$, and $NaHCO_3$, our ability to observe the stable ice III (or ice II below 238 K) transition and determine the cenotectic was confounded by the apparent emergence of high-pressure intermediates, some of which we anticipate may represent previously unknown phases. P-T coordinates measured for each of these systems under different cooling protocols (see "Methods") are shown individually in Fig. 2c (with additional runs shown in Supplementary Note 3), and suggest that isochoric freezing and melting may prove an effective technique for screening for new phases in low-temperature aqueous systems.

For the 23.3% NaCl solution, which begins in a eutectic configuration comprised of liquid, ice Ih, and hydrohalite ($NaCl\cdot 2H_2O$, or SC2), an intermediate phase transition was observed at approximately 237 K / 140 MPa, remarkably close to the temperature at which Journaux and colleagues estimate the recently discovered hyper-hydrate

$2NaCl\cdot 17H_2O$ (SC8.5) may form[7]. This second phase configuration continued without interruption to approximately 215 MPa across trials but achieved this pressure at between 216 and 224 K depending on the run. The P-T coordinates of the initial eutectic configuration agree well with previous data from Kanno & Angell (black triangles in Fig. 2c), who observed high-pressure eutectic phenomena whilst studying pressurized homogeneous ice nucleation via differential thermal analysis in emulsified $H_2O$-NaCl samples[4]. They do not report an additional phase transition, however, which is likely a kinetic consequence of both emulsification and miniscule sample sizes (0.04–0.08 mL), which they deploy in order to hinder heterogeneous nucleation of new phases. The large sample sizes (here 5.33 mL) deployable in isochoric freezing and melting may provide a distinct advantage in this regard, helping to ensure timely heterogeneous nucleation of new phases.

In 6.15% $NaHCO_3$ solution, which begins in a eutectic configuration comprised of liquid, ice Ih, and pure $NaHCO_3$, a phase transition was observed at approximately 255 K / 130 MPa, suggesting the existence of what may be the first reported hydrate for $NaHCO_3$, at high-pressure or otherwise. Chamber-to-chamber P-T results grow less consistent after this first phase transition, with two runs eventually unifying in pressure at 210 MPa (albeit at 232 K and 238 K), and the third taking a much shallower approach in temperature. The transition at 210 MPa suggests that another phase transition may occur in this 232–238 K temperature range, be it within the salt-bearing solid phase or the ice phase. Future work should address the structure and stability range of these (potentially multiple) new hydrates and may provide fundamental new insight into the hydration dynamics of NaHCO3.

Finally, in 21.6% $MgCl_2$ solution, which begins in a eutectic configuration comprised of liquid, ice Ih, and $MgCl_2\cdot 12H_2O$, discrete phase transitions were observed in multiple runs at approximately 233 K / 75 MPa and 223 K / 155 MPa, after which the behaviors of the different samples diverged significantly. $MgCl_2$ is known for its diverse array of stable hydrate phases, with hydration numbers up to 12 observed at ambient pressure[8]. However, to our knowledge, the only hydrate reported to exist in equilibrium with ice Ih is MgCl2-12H2O, suggesting that the phase transitions observed here may represent previously unrealized phases.

## Discussion

### Role of water-ice stability in prescribing the cenotectic

Our measurements provide several key insights into the evolution of aqueous systems with pressure. Firstly, it appears that for binary salt systems that do not undergo transitions to a salt-bearing solid phase at high pressures, the P-T coordinates of the cenotectic are dictated nigh-exclusively by the thermodynamics of water/ice. As shown in Fig. 2b, the P-T evolution of each eutectic configuration from its 0.1 MPa eutectic point mirrors the P-T evolution of ice Ih and liquid water from 273.15 K, consistently producing the ice III transition (which here marks the cenotectic) at approximately 22 K beneath the melting point of ice Ih, and at approximately 210 MPa. This is a key finding for analysis of binary aqueous salt systems of relevance to planetary science, cryopreservation, etc., providing a general thermodynamic limit on the presence of the stable equilibrium liquid. We

**Table 1 | Measured multiphase P-T equilibria for solutions exhibiting monotonic P-T eutectic curves**

| SoluteSalt | Initial brine concentration (Eutectic concentration at 0.1 MPa; wt %) | Eutectic Temperature at 0.1 MPa (K) | Cenotectic Temperature (K) | The temperature difference between Eutectic at 0.1 MPa and Cenotectic (K) | Cenotectic Pressure (MPa) |
|---|---|---|---|---|---|
| $Na_2CO_3$ | 5.88 | 270.77 (0.14) | 247.11 (0.48) | 23.66 | 217.41 (1.61) |
| KCl | 19.50 | 262.41 (0.16) | 238.90 (0.52) | 23.52 | 213.33 (1.58) |
| $MgSO_4$ | 17.30 | 269.29 (0.10) | 247.56 (0.34) | 21.75 | 207.31 (0.94) |
| Urea | 32.80 | 261.74 (0.18) | 240.50 (0.53) | 21.21 | 210.38 (1.24) |
| $Na_2SO_4$ | 4.15 | 272.99 (0.13) | 248.01 (0.54) | 23.98 | 214.46 (1.80) |

furthermore hypothesize that if no new stable hydrate phases are present at high pressures in *any* given aqueous salt solution, be it a binary or many-component system, this 22 K rule of thumb should hold true for the temperature difference between the 0.1 MPa univariant eutectic and the cenotectic. Following this logic, as the ice Ih-III-II triple point is located around 238 K and 210 MPa, systems with 0.1 MPa eutectics > 260 K (238 + 22) are predicted to possess a cenotectic at the ice Ih-ice III-liquid triple point, and systems with 0.1 MPa eutectics < 260 K are predicted to possess a cenotectic at the ice Ih-ice II-liquid triple point.

For clarity, in Fig. 3, we have extended the conceptual phase diagram presented in Fig. 1 to include prototypical examples of solute solubility surfaces that may correspond to these differing cenotectics, organizing them by molar solubility. Examples of low solubility (e.g., aqueous $Na_2SO_4$ or $MgSO_4$) and medium solubility (e.g., aqueous KCl or Urea) cenotectics are represented within the experimental data in Fig. 2a, b, and marked conceptually in Fig. 3 by $\kappa_1$ and $\kappa$. Examples of high solubility cenotectics (occurring at the ice ih-ice II-liquid triple point and marked by $\kappa_3$) have not been experimentally confirmed in this work, but we suggest that aqueous perchlorates may be good candidates, as they often possess very low 0.1 MPa eutectic temperatures (e.g., 236 K for $NaClO_4$ and 206 K for $MgClO_4$). For aqueous NaCl and $MgCl_2$, while we observe signs of new hydrate species at high pressures that may increase the temperature of the cenotectic, it appears likely that they will yet fall beneath 238 K (along the Ih-ice II-liquid triple point), considering the trends of their eutectic curves (Fig. 2c).

We suspect that the general dominance of water-ice thermodynamics over the trajectory of the univariant (here binary eutectic) configuration is a function of the uniquely high phase volume difference between liquid water and ice Ih, which renders them significantly more sensitive to pressure (in this temperature range) than the salt-bearing solid phases. We emphasize, however, that in hypothetical aqueous systems wherein the salt-bearing solid phase boundaries may vary significantly with pressure over the same ~ 210 MPa range relevant to ice Ih, this 22 K rule of thumb need not necessarily hold.

## Intermediate salt-based phase transitions along the eutectic curve

As shown in Fig. 2c, three binary systems that produced intermediate phase transitions within approximately 22 K of their 0.1 MPa eutectic points proceeded to diverge from the eutectic trajectories of their monotonic counterparts.

Unlike in the previous solutions, these binaries showed notable sensitivity to the cooling and warming protocol employed and divergence amongst like samples, producing anomalous P-T curves. At equilibrium, a stable phase will have the highest melting temperature at a fixed pressure and composition (or conversely, the highest pressure at a fixed temperature and composition). Proceeding along the eutectic curve (toward lower temperatures and higher pressures), if reaching the range of stability of a new hydrate, one should expect a discontinuity of slope in the P-T trajectory, with the value of the Clapeyron slope becoming more negative (steeper). Here we observe the opposite, with the slope becoming more positive (less steep) after discontinuity. This suggests that possible metastable phases of hydrates may have formed during cooling and that during warming (when step-wise P-T data is recorded), these various phases may reach their metastable melting point(s) below the cenotectic temperature, therefore creating the complex isochoric path we report here.

For example, for the $H_2O$-NaCl system, several new hydrates have recently been reported, including NaCl·8.5($H_2O$) (SC8.5), which possesses a melting point measured at 240 K at 380 MPa, and an extrapolated 0.1 MPa metastable melting point at 235K[7]. Along the NaCl(aq) isochoric P-T eutectic curve, we see a discontinuity of slope at approximately 140 MPa and 238 K. This is where one would expect SC8.5 to cross its metastable melting curve at that pressure. This illustrates how a hydrate, stable at higher pressure, can leave a clear trace as a metastable phase in an isochoric experiment. Accordingly, this suggests that the other systems with anomalous isochoric paths possess other undiscovered hydrates, potentially stable at higher pressures. This supports the hypothesis of pressure-induced hydrate structural diversity (hyper-hydration) from Journaux et al.[7], and we expect one or multiple new hydrates to be stable at higher pressures in $MgCl_2$ and $NaHCO_3$. This underlines the power of the isochoric method

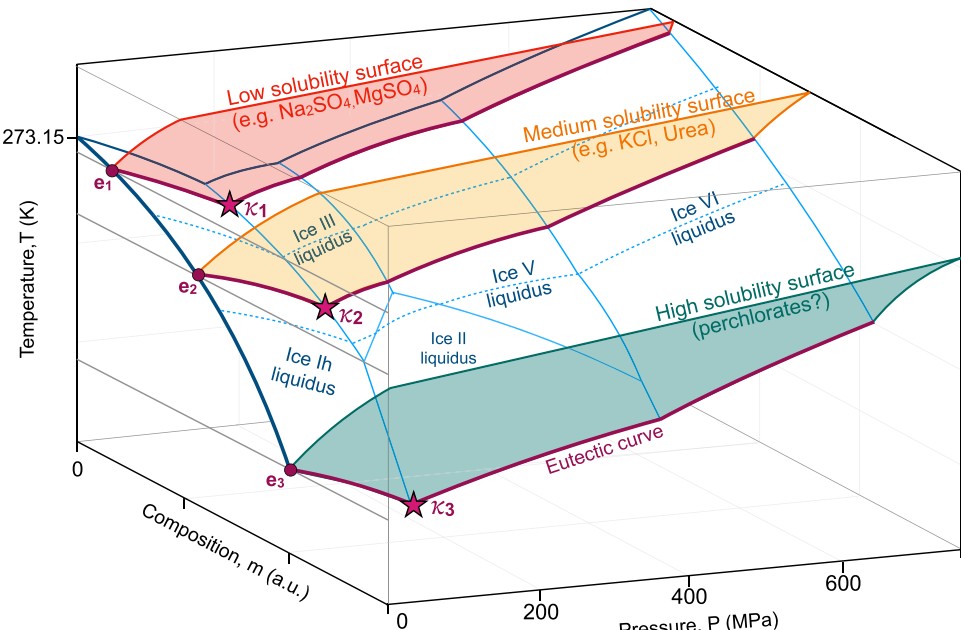

**Fig. 3 | Conceptual pressure-temperature-concentration phase diagram depicting binary solutions of water and generic salt-like solutes of varying solubilities.** A generic low solubility surface is shown in red, a generic medium solubility surface is shown in yellow, and a generic high solubility surface is shown in green. The 0.1 MPa eutectic points corresponding to each surface are marked by ($e_1$, $e_2$, $e_3$), and the cenotectics are marked by ($\kappa_1$, $\kappa_2$, $\kappa_3$), where increasing subscript indices correspond to increasing solubility. All other markings retain the meanings indicated in Fig. 1.

to rapidly explore the fundamental thermodynamics and physical chemistry of these systems, and future experiments should investigate the high-pressure stability and structure of hydrates in those systems.

Finally, it should be noted that the formation of metastable intermediates confounds our ability to directly measure the stable cenotectic temperature in these chemical systems. This presents an interesting physical paradox: new hydrates encountered along the eutectic curve that originates at 0.1 MPa must definitionally be stable at higher pressures than the eutectic P-T curve would provide; but as observed, they may produce metastable configurations at lower pressures. This implies that the formation of new stable hydrates, at thermodynamic equilibrium, will increase the cenotectic temperature, ushering the system to pressures at which ice II or III may form at higher temperatures (less than the 21 K rule of thumb) and limiting the stability range of the liquid. Simultaneously, the formation of metastable new hydrates will decrease the cenotectic temperature, extending the temperature range over which liquid may exist by tens of degrees or more (Fig. 2c). Additional investigation into kinetic aspects of these systems (further cooling rate sensitivities, effects of different annealing temperatures, effects of spontaneous or stimulated nucleation temperatures, etc.) may yield deeper insight into their potential metastability.

## Implications for materials thermodynamics and generalized definition of the cenotectic

For any thermodynamic system, under the influence of any arbitrary selection of physical or chemical thermodynamic forces (pressure, chemical potential, etc.) acting at arbitrary intensities, there exists a temperature limit to the stability of the liquid phase, which we here dub the cenotectic. For systems free of the influence of non-simple modes of thermodynamic work (e.g., magnetic work, electronic work, stress-strain work, surface work, etc.), this limit will be a function of pressure and $c - 1$ compositional variables, wherein $c$ is the number of chemical components. For systems under the influence of the more exotic forms of thermodynamic work mentioned above, the cenotectic will depend further on one additional variable for each contributing work mode. As such, this cenotectic limit is a fundamental material property of the system, yet, outside of chemically pure systems, it has seldom (if ever) been precisely characterized.

In this work, we have measured the cenotectic for a variety of binary salty aqueous systems. However, these systems represent just one of several phenomenological profiles that the cenotectic may possess. Because mechanical (PV) work affects all thermodynamic systems, the cenotectic will always occur at the intersection of two univariant phase lines with opposite-signed Clapeyron (dP/dT) slopes, regardless of chemical complexity or the role of other thermodynamic fields. For the systems tested here, because water produces a solid phase (ice Ih) less dense than the liquid (thereby providing a negative Clapeyron slope along its liquidus curve), and because the eutectic featuring ice Ih is the lowest-temperature eutectic in the atmospheric pressure phase diagram, the cenotectic logically falls at the intersection of two univariant liquid-solid-solid equilibrium lines, one involving ice Ih (plus the salt-bearing solid phase) and the other involving an ice phase (II or III) denser than the liquid (plus the salt-bearing solid phase). This reasoning can be applied equally to other systems with negative Clapeyron slope melting curves, for which the cenotectic can fall along the solid-solid transition to the denser phase (which is usually quasi-isobaric). We, therefore, predict that the cenotectic pressure can be expected to be approximately 11-12 GPa for many silicon solutions, 1.5 GPa for many gallium solutions, and 5–10 GPa for many carbon solutions.

However, for most binary material systems, which do not possess a solid phase less dense than the liquid along the eutectic curve, the negative-Clapeyron phase line leading into the phase intersection marked by the cenotectic will generally include liquid-vapor-solid equilibrium, instead of liquid-solid-solid equilibrium. Likewise, in aqueous solutions for which the lowest-temperature eutectic at atmospheric pressure does not involve ice Ih, such as methanol[9], DMSO[10], etc., the cenotectic may also occur at the intersection of a liquid-vapor-solid (negative Clapeyron slope) line and a liquid-solid-solid (positive Clapeyron slope).

Generally, the cenotectic point of a system will behave one of two ways, depending upon the densities of the phases involved. If the lowest-temperature eutectic at atmospheric pressure involves a phase less dense than the liquid, the cenotectic will likely occur at increased pressures, and generally at the intersection of negative-Clapeyron and positive-Clapeyron liquid-solid-solid equilibrium lines. This is the case applicable to all of the systems studied here, and the case we suggest should be broadly applicable to aqueous salt systems writ large. Alternatively, if the lowest-temperature eutectic at atmospheric pressure involves only phases that are denser than the liquid, the cenotectic will likely occur at decreased pressures, and generally at the intersection of a positive-Clapeyron liquid-solid-solid and a negative-Clapeyron liquid-vapor-solid line. This appears to be the case, for example in the water-methanol system[9], for which the lowest temperature eutectic at 0.1 MPa does not include ice Ih, but instead solid methanol and its monohydrate. Note, however, that for systems with multiple eutectics at similar temperatures (such as water-DMSO[10]), the specific pressure-dependences of the solid phases involved may affect which 0.1 MPa eutectic the cenotectic will originate from, and a priori determination of the phases involved may not be feasible.

It must also be noted that the number of phases present in the univariant phase lines that intersect to form the cenotectic will increase with the number of chemical components in the system, per the classical Gibbs Phase Rule, and with the number of additional thermodynamic fields (strain, magnetic, electric, etc.) acting upon the system, per the Generalized Gibbs Phase rule[11], leaving many exciting multiphase configurations to characterize across varying thermodynamic systems.

We suggest that establishment and exploration (both experimental and theoretical) of the cenotectic concept is vital to a rigorous understanding of the thermodynamics of modern materials systems, which exist under increasingly complex chemical-thermodynamic conditions and produce increasingly rich multi-phase equilibria[12–15]. For the benefit of the modern student of thermodynamics, we thus conclude our thermodynamic discussion with a formal definition of the cenotectic.

The cenotectic is the invariant point occurring at the lowest temperature at which the liquid phase, for any value of concentration, pressure, or other thermodynamic forces acting on the system, remains in stable equilibrium.

Additional discussion (and a mathematical description) of this definition within the context of Gibbs Phase Rule is provided in Supplementary Note 5.

## Implications for planetary science

The cenotectic concept has fascinating applications in planetary science, especially for cold water-rich worlds like the icy moons of the outer solar system and ocean exoplanets. In icy worlds such as Europa, Enceladus, Titan, Ganymede, Ceres, Pluto, and potentially moons of Uranus Ariel, Umbriel, Titania, and Oberon[16–18], the eutectic behavior controls the stability of brines in the icy crust. This particularly affects the circulation of salty fluids and the formation of mushy layers (Fig. 3) at the uppermost ocean boundaries, which may provide habitable niches for potential extraterrestrial life[19–21]. With increasing pressure deeper within the ice Ih crust, the temperature of the eutectic will decrease (Fig. 4). Several studies in recent years have underlined the importance of brine percolation and brine connectivity in vertical transport through the ice crust under chaos regions[22], double ridge formations[23], or impacts and cryo-volcanism[20,24,25]. The extent to which brines formed near the surface can percolate through the ice shell by various mechanisms and act as nutrient sources for the underlying

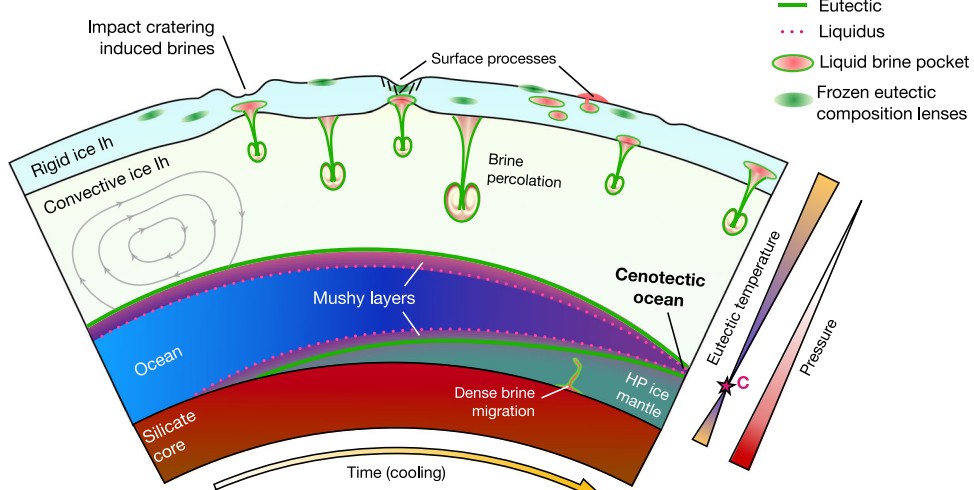

**Fig. 4 | Schematic interior structure of an icy moon and the various geological processes involving the eutectic and cenotectic.** These processes include but are not limited to brine pocket migrations, percolation, possible eruptions (eutectic), and freezing of the hydrosphere from above (ice Ih) and below (high-pressure ices) that results in the cenotectic ocean. Ice Ih layer relative thickness is exaggerated for clarity.

oceanic habitat is *in-fine* controlled by the eutectic coordinates (Fig. 4). The present work allows us to demonstrate that for a pressure of the bottom of the icy shell of 50 MPa[17,26] eutectic temperature of aqueous systems can be depressed by up to 4-5 K compared to its 0.1 MPa value. This difference is significant and could enhance fluid circulation in thick ice crusts[22,27], further facilitating vertical transport of oxidants/ nutrients from the surface to the ocean. The high-pressure behavior of the eutectic is also relevant to the base of the ice shell, especially in controlling the formation, thickness, and chemical gradients of ocean interface mush[28,29] (Fig. 4), and possible briny habitats[19].

The behavior of the eutectic curve at pressures higher than the cenotectic, in equilibrium with high-pressure ice polymorphs, remains to be studied with other techniques. Nonetheless, a reasonable assumption is to expect a similar behavior to that at low pressure, i.e., following the overall shape of the high-pressure ice melting curves with a given ΔT, if no high-pressure hydrates are present (Fig. 2b). For aqueous salt systems with cenotectic temperatures colder than 238 K, ice II should form, and the eutectic will follow the ice II-aqueous fluid melting line, which remains to be measured experimentally for relevant solutions. To our knowledge, ice II melting curves are only available for a few select solutes, including aqueous ammonia and aqueous methanol[9,30].

For large icy ocean worlds with high-pressure ices (e.g., Ganymede, Titan, Callisto) and ocean exoplanets[26], the coordinates of the eutectic curve will also control the circulation of metamorphic fluids or fluids from hydrothermalism through the high-pressure ice mantle, and dictate the possibility of vertical exchange of nutrients from the rocky core towards the ocean (Fig. 4). These processes and implications for geological evolution and habitability of these large moons are discussed in detail in references[31–36].

Finally, the cenotectic plays a central role in the "endgame" of planetary oceans. As large water-rich planetary bodies cool over geologic timescales or with loss of internal heating such as tidal dissipation or radiogenic heating, their oceans will gradually freeze from top to bottom (Fig. 4), until complete solidification is achieved. This effect is particularly interesting in the case of large icy moons like Ganymede, Callisto, and Titan, but also for cold ocean exoplanets like Trappist 1e-g and water-rich rogue exoplanets. Formation of a cenotectic ocean (i.e., an ocean existing at the coldest temperature at which the aqueous liquid will remain stable) could happen in the case that the aqueous solution stays buoyant compared to high-pressure ices, which depends on the solute molar mass and the composition of the eutectic. Otherwise, the brine could be transported downward and may form

dense basal oceans, as predicted for NaCl and MgSO$_4$ brines[36–38]. This important density inversion effect, which could control the distribution of aqueous reservoirs in water-rich planetary bodies, requires further investigation into the composition and density of aqueous systems along the eutectic and at the cenotectic. In the following discussion we assume that the brine at the cenotectic composition remains buoyant compared to high-pressure ices, as could be expected with the presence of ammonia, for example. We also assume that all of the chemical constituents dominating the composition of these icy oceans possess high-pressure cenotectics, like the salt systems studied here.

For cold water-rich planetary bodies, the last remnant aqueous liquid layer will occur at the depth corresponding to the pressure of the cenotectic ($P_\kappa$), as shown in the present study, to be around 210 MPa. In Fig. 5, this cenotectic depth $z_\kappa$ of the last ocean is presented as a function of gravity, and discrete values are reported in Table 2. Cenotectic depth is estimated using the formula $z_\kappa = P_\kappa / (\rho \cdot g)$, with the density $\rho$ of the overlying ice taken as 930 kg/m$^3$ (the average value for an adiabatic thermal profile in an ice Ih crust is around 250 K[26]), and $g$ taken as the surface gravitational acceleration of the planetary body[39]. Bodies with smaller gravity will have their last remaining aqueous liquid at greater depth, over 150 km, for all large icy moons in our solar system. Small objects like Ceres, Pluto, and others are omitted here, as the calculated depth of the cenotectic is shallower than the estimated thickness of their hydrospheres. For Europa, the depth of the cenotectic roughly corresponds to the estimated thickness of its hydrosphere. For larger icy moons, the final ocean will be sandwiched between layers of ice Ih and ice III or ice II, depending on the precise composition of the remaining aqueous liquid. Interestingly, the depth of the cenotectic also provides the absolute maximum upper limit for the ice crust thickness if an ocean is present, which is 172 km for Titan, and 158 km for Ganymede. For larger bodies with larger gravitational acceleration, such as water-rich exoplanets, the depth of the final ocean will be shallower, with a depth ranging from 28 km for TRAPPIST-1 e to 9.7 km for LHS 1140 b, implying much thinner maximum ice Ih crust (< 30 km) for cold ocean super earth.

## Future work and perspective on the cenotectic
The cenotectic, or the invariant point defining the lowest temperature at which a liquid remains stable under any possible values of concentration, pressure, and other thermodynamic parameters affecting

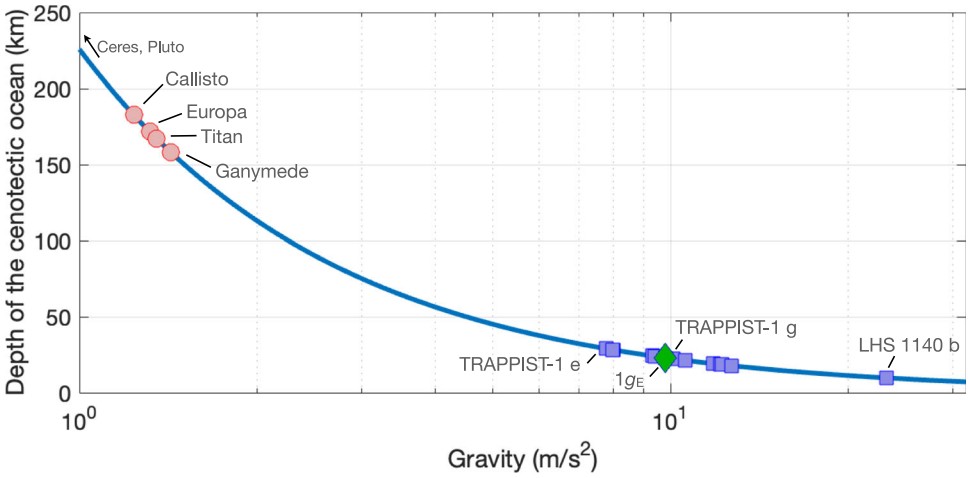

**Fig. 5 | Evolution of the final depth of the cenotectic ocean (cenotectic depth) for different planetary bodies from our solar system and ocean exoplanet candidates.** The cenotectic ocean is the ocean occurring at the depth where the last liquid layer remains stable during the freezing of the hydrosphere over the course of geological time. Surface gravity values are taken from Ojha et al.[39].

the system, is an important concept in the material characterization of complex multiphase systems. As the thermodynamic and compositional parameter spaces of interest to the research community grow ever larger, identifying globally limiting behaviors becomes increasingly essential, and we suggest that the cenotectic concept may help to establish definite and fundamental limits on liquid phase equilibria. With that in mind, future research should aim not only to measure the cenotectic limits of many other chemical systems (including those aqueous systems relevant to potential high-pressure, low-temperature applications such as cryopreservation[40–42]) but to further refine and explore the concept itself, with the goal of an ever deeper understanding of the physical limits of material systems.

## Methods

### Isochoric freezing and melting process

In order to identify the temperature limit of liquid stability in the aqueous solutions explored herein, we extended the isochoric freezing and melting technique reported previously by Chang et al.[1] to deeper temperatures. In brief, 5.33 mL aqueous samples mixed at the atmospheric-pressure eutectic concentration were confined absent air within a custom Al7075 isochoric chamber fitted with a high-pressure transducer (ESI Technology Inc). By constraining the total volume of the binary system and allowing the pressure to vary freely, we create a thermodynamic environment in which the 3-phase equilibrium present in the eutectic configuration is prescribed by a single intensive thermodynamic degree of freedom. As such, temperature and pressure are coupled within the chamber, and simple control of the temperature enables simultaneous measurement of the equilibrium temperature and pressure. Further experimental details and chamber schematics are provided in Supplementary Notes 1–3, and described below, with full experimental procedures provided to ensure reproducibility.

### Freezing process

Two high-accuracy recirculating cooling baths were employed for isochoric testing, a PolyScience AP15R-40 with a minimum working temperature of 243.15 K and a Fluke Calibration 7380 with a minimum working temperature of 193.15 K. The latter was utilized for solutions with phase transitions lower than 243.15 K (including all those exhibiting intermediate high-pressure hydrate phases), and the former for all others. For each solution, three replicate samples in identical chambers were studied, and the chambers were submerged completely within the cooling bath fluid to ensure uniformity of

temperature. More details on the cooling methodology and equipment are available in Supplementary Note 1, and a sample cooling profile is shown in Supplementary Note 2.

For solutions without intermediate high-pressure hydrates phases (i.e., those with monotonic P-T trajectories), the chambers are plunged into a bath pre-cooled to at least 30 K beneath the 0.1 MPa eutectic temperature of the solution, monitored until the ice III/II transition is observed (indicated upon cooling by a precipitous collapse in pressure to approximately 210 MPa; see Supplementary Fig. S2), and then allowed to further equilibrate until the pressure stabilizes to < 0.1 MPa/min variation (typically 1-2 h).

For solutions found to host intermediate high-pressure hydrate phases ($NaCl$, $NaHCO_3$, and $MgCl_2$, i.e., those with non-monotonic P-T trajectories), contrary to their less phase-diverse counterparts, preliminary testing showed broad varieties in behavior sample-to-sample,

**Table 2 | Cenotectic depth for various relevant icy moons of our solar system and ocean exoplanet candidates with effective surface temperatures below 273 K**

| Object | Surface Gravity (m/s²) | Depth of the cenotectic (km) (i.e., last ocean) |
|---|---|---|
| Callisto | 1.24 | 182 |
| Titan | 1.35 | 167 |
| Europa | 1.32 | 172 |
| Ganymede | 1.43 | 158 |
| TRAPPIST-1 e | 7.98 | 28 |
| TRAPPIST-1 f | 9.41 | 24 |
| $1g_E$ exoplanet | 9.81 | 23 |
| Teegarden's Star c | 10.04 | 22.5 |
| TRAPPIST-1 g | 10.12 | 22.3 |
| Proxima Cen b | 10.6 | 21.3 |
| GJ 667 C f | 11.82 | 19.1 |
| GJ 667 C e | 11.82 | 19.1 |
| GJ 1061 d | 12.14 | 18.6 |
| Keppler-186 f | 12.23 | 18.5 |
| Keppler 2-18b | 12.43 | 18.2 |
| Keppler-442 b | 12.68 | 17.8 |
| Keppler-1229 b | 12.68 | 17.8 |
| LHS 1140 b | 23.22 | 9.7 |

leading us to believe that metastable configurations were being produced and that a more gradual cooling approach may be necessary. As such, for the protocol identified as Slow Cooling in Fig. 2c, the chambers were pre-cooled only to 273.15 K, and the bath temperature was then decreased by 1 K every 10 min. The protocol identified as Fast Cooling follows the approach described above for solutions without high-pressure intermediates, wherein the chamber is submerged directly into liquid pre-cooled to the minimum temperature of the run, with the exception that pressure equilibration after the ice III/II transition typically required 6–8 h. In order to demonstrate the hallmark dependence of metastable configurations on thermal history, a combined slow-fast cooling approach was also tested, which incorporated a pseudo-annealing logic. In this process, the temperature was incrementally lowered at a rate of 1 K per 10 min until the intermediate hydrate was detected, then raised 5 K above the observed transition temperature and cooled once more at a rate of 0.5 K/min to the cenotectic point.

See Supplementary Note 2 for example time-series data on the freezing process, and Supplementary Note 3 for additional data on solutions with intermediate hydrates.

### Melting process

For all solutions, after completion of the cooling process, the system is heated in discrete increments of 0.5 K, with time increments chosen based on the heat transfer characteristics of each bath to produce steady steady-state P-T data by the end of each step (at which point the data is sampled), as indicated by a < 0.1 MPa/min evolution of the pressure. Additional equipment details are available in Supplementary Notes 1 and 2, and an example cooling-warming profile is shown in Supplementary Fig. S2. The steady pressure and temperature are recorded at the end of each step, and this pressure-temperature data provides the P-T curves shown in Fig. 2.

### Eutectic and cenotectic point identification

In order to extract more precise cenotectic points from the P-T data shown in Fig. 2a, second-order polynomials were fit to the data above and below the apparent cenotectic temperature (as indicated by the discontinuity in the slope of the P-T curve) using the polyfit() function in MATLAB 2023b, and their intersection provided the cenotectic temperatures/pressures reported in Table 1.

Similarly, following the approach of Chang et al.[1], the intersection of the higher-temperature fitted polynomial curve with the 0.1 MPa isobar identifies the atmospheric pressure eutectic temperature of the system.

Propagating uncertainty analysis was performed to account for uncertainty in both the experimental setup and the fitting procedure and is described in detail in Supplementary Note 4.

## Data availability

All data reported herein are available in tabular form in the Source Data. Any additional information sought may be requested from the corresponding authors. Source data are provided in this paper.

## Code availability

All code employed for the analysis of data herein is available upon request to the corresponding authors. We will note, however, that only standard MATLAB 2023b curve fitting functions were employed, and no custom scripts were developed for analysis.

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

## Acknowledgements

The authors would like to thank Claire Mieher of the University of Southern California for her linguistic expertise, and suggestion of the term cenotectic. The authors would also like to thank Dr. Thomas Driesner for his invaluable feedback on the manuscript and figures. B.J. acknowledges the financial support provided by NSF through "The Chemistry of Aqueous Carbonic Fluids in Subduction" grant (HD1WMN6945W6), the NASA Astrobiology Institute through the Titan and Beyond node (17-NAI82-17), and the NASA Precursor Science Investigations for Europa grant (22-PSIE22_2-0024). A.Z. and M.P.P. acknowledge funding from the NSF Engineering Research Center for Advanced Technologies for Preservation of Biological Systems (ATP-Bio) NSF EEC #1941543.

## Author contributions

M.P.P. and B.J. developed the cenotectic concept and conceived the study. A.Z. performed all experiments and data analysis and contributed key conceptual insights. M.P.P. supervised all experimental work and analysis. B.J. performed the planetary science analyses and supervised the analysis of all experimental data. All authors contributed to the writing and review of the manuscript.

## Competing interests

The authors declare no competing interests.
