## [Transparent Peer Review file · Nature Communications]

On the equilibrium limit of liquid stability in pressurized aqueous systems

Corresponding Author: Professor Matthew Powell-Palm

Version 0:

Reviewer comments:

Reviewer #1

(Remarks to the Author)

The manuscript "On the equilibrium limit of liquid stability in pressurized aqueous systems" presents the results of an experimental study, involving the isochoric freezing of binary aqueous brines, which demonstrates the existence of a "cenotectic" defined as the "invariant point occurring at the lowest temperature at which the liquid phase, for any value of concentration, pressure, or other thermodynamic forces acting on the system, remains in stable equilibrium". Although this work was limited to binary aqueous brines, the manuscript highlights that the results can be extended more generally to other systems, where the location of the cenotectic will be governed by intersection of phase equilibrium lines with opposite Clapeyron slopes. The existence of a cenotectic is demonstrated to be significant to icy ocean worlds since it governs the depth at which the final remnant of a liquid ocean solidifies. Implications for cryopreservation are also highlighted.

The manuscript represents an important contribution to the scientific literature and introduces a key concept that has implications across a diverse range of fields. The results are compelling, and the methodology is well-founded. However, there are a few areas where the manuscript could be improved.

Specific recommendations for minor edits are included in the attachment, but more significant comments are summarized here. We recommend the manuscript be published after these comments are sufficiently addressed.

-Figure 1:

- Please label the concentration axis and include the appropriate limits. Currently the axis value is 0 at both ends (corresponding to the 0 value for temperature and pressure).
- Since the variables P-T-x are used throughout the manuscript, I recommend including them in the label, Example: Pressure, P (GPa)
- Please define m_e , m_c , P_c in the figure caption
- Consider including a 2D cutout (or multiple 2D cutouts) of the figure intersecting at point C
- It's not clear that m_e is different from m_c given your experiments, please elaborate on this point somewhere in the manuscript

-Discontinuity:

- The term discontinuity is used to describe the conditions at which the cenotectic is reached. This is not formally a discontinuity in the eutectic curve itself but perhaps a "kink" or more explicitly a "discontinuity in the slope"?

-Table 1:

- Consider emphasizing that the initial concentration is the eutectic concentration at 1 atm, since it's not immediately apparent from the table.
- For clarity, please write out "Difference between Eutectic Temperature at 0.1 MPa and cenotectic temperature" instead of or in addition to ΔT

-Figure 2:

- Please label plots with a, b, c
- In addition to ΔT , please write out "Difference between Eutectic and Cenotectic Temperature"

- Please include variables along with axis labels, Example: Temperature, T (K)

-Table 2:

- The gravitational accelerations and therefore depth of the cenotectic for Titan and Europa appear to be switched
- Consider including the source for the gravitational acceleration, particularly for the exoplanets

-Figure 4:

- The gravitational accelerations and therefore depth of the cenotectic for Titan and Europa appear to be switched

-Terminology

One recurring but minor issue throughout the manuscript is the lack of consistency and precision in terminology:

- “eutectic line” vs. “eutectic curve” vs. “P-T curve” vs. “P-T trajectory” vs. “eutectic pressure-temperature profile”. To minimize confusion, I recommend adopting a single term for the “eutectic curve” and using it throughout the manuscript.
- If using the term “P-T coordinates” specify if these are “P-T coordinates of the eutectic points” or “P-T coordinates of the cenotectic point”.
- Similarly for atmospheric pressure 1 atm, 1 bar, 0.1 MPa?
- T-P vs P-T (I recommend P-T)
- P-T-X vs. P-T-x

Reviewer #2

(Remarks to the Author)

The manuscript “On the equilibrium limit of liquid stability in pressurized aqueous systems” presents the results of an experimental study, involving the isochoric freezing of binary aqueous brines, which demonstrates the existence of a “cenotectic” defined as the “invariant point occurring at the lowest temperature at which the liquid phase, for any value of concentration, pressure, or other thermodynamic forces acting on the system, remains in stable equilibrium”. Although this work was limited to binary aqueous brines, the manuscript highlights that the results can be extended more generally to other systems, where the location of the cenotectic will be governed by intersection of phase equilibrium lines with opposite Clapeyron slopes. The existence of a cenotectic is demonstrated to be significant to icy ocean worlds since it governs the depth at which the final remnant of a liquid ocean solidifies. Implications for cryopreservation are also highlighted.

The manuscript represents an important contribution to the scientific literature and introduces a key concept that has implications across a diverse range of fields. The results are compelling, and the methodology is well-founded. However, there are a few areas where the manuscript could be improved.

Specific recommendations for minor edits are included in the attachment, but more significant comments are summarized here. I recommend the manuscript be published after these comments are sufficiently addressed.

Figure 1

- Please label the concentration axis and include the appropriate limits. Currently the axis value is 0 at both ends (corresponding to the 0 value for temperature and pressure).
- Since the variables P-T-x are used throughout the manuscript, I recommend including them in the label, Example: Pressure, P (GPa)
- Please define m_e , m_c , P_c in the figure caption
- Consider including a 2D cutout (or multiple 2D cutouts) of the figure intersecting at point C
- It’s not clear that m_e is different from m_c given your experiments, please elaborate on this point somewhere in the manuscript

Discontinuity

- The term discontinuity is used to describe the conditions at which the cenotectic is reached. This is not formally a discontinuity in the eutectic curve itself but perhaps a “kink” or more explicitly a “discontinuity in the slope”?

Table 1

- Consider emphasizing that the initial concentration is the eutectic concentration at 1 atm, since it’s not immediately apparent from the table.
- For clarity, please write out “Difference between Eutectic Temperature at 0.1 MPa and cenotectic temperature” instead of or in addition to ΔT

Figure 2

- Please label plots with a, b, c
- In addition to ΔT , please write out “Difference between Eutectic and Cenotectic Temperature”
- Please include variables along with axis labels, Example: Temperature, T (K)

Table 2

- The gravitational accelerations and therefore depth of the cenotectic for Titan and Europa appear to be switched
- Consider including the source for the gravitational acceleration, particularly for the exoplanets

Figure 4

- The gravitational accelerations and therefore depth of the cenotectic for Titan and Europa appear to be switched

Terminology

One recurring but minor issue throughout the manuscript is the lack of consistency and precision in terminology.

- “eutectic line” vs. “eutectic curve” vs. “P-T curve” vs. “P-T trajectory” vs. “eutectic pressure-temperature profile”. To minimize confusion, I recommend adopting a single term for the “eutectic curve” and using it throughout the manuscript.
- If using the term “P-T coordinates” specify if these are “P-T coordinates of the eutectic points” or “P-T coordinates of the cenotectic point”.
- Similarly for atmospheric pressure 1 atm, 1 bar, 0.1 MPa?
- T-P vs P-T (I recommend P-T)
- P-T-X vs. P-T-x

Reviewer #3

(Remarks to the Author)

Zarriz et al. report first-of-their-kind experimental data on low-temperature (T) / high pressure (P) phase relations of aqueous salt systems, namely between the different ice polymorphs and salt-bearing solids. They claim to determine the T-P coordinates of the low-temperature endpoint of the cotectic between ice and a solute-bearing solid, which they name the “cenotectic” and suggest that this is the lowest possible temperature at which the respective aqueous salt solution can stably exist. They then explore implications for thermodynamics of such systems and applications to “icy moon” environments.

The experimental findings as such are very interesting and novel. Yet, I had a very hard time preparing this review as the organization of the manuscript is poor. In my opinion it requires a major re-write / thorough major revisions, followed by another round of reviewing, before it can be considered for publication in a leading peer-reviewed journal. Namely, a clear documentation of the experimental approach and its likely first-order interpretation is largely absent although it would be absolutely essential to make the text more readable and the work reproducible. Regarding discussion of the implications a much stronger and clearer connection to the experimental results should be made.

Some of my concerns in detail are:

- Perhaps my biggest criticism is that the authors do not clearly (and not even very briefly) document how they did the experiments and that the remaining descriptions and wording in the text are really misleading in this respect. If I understood it correctly – and it is really unacceptable that one has to read a previous publication (Chang et al.) to get to this point – the experiments are actually NOT freezing experiments but rather melting / heating experiments at very low temperatures. The experimental charge is first frozen and then slowly heated and phase transitions are recorded as changes of temperature/pressure slopes upon heating in the isochoric system. Unfortunately, this is never clearly spelled out (a “Methods” or “Experimental” section would have helped). In fact, much of what should be part of such an early-on description of what was done and observed comes as interpretation in the discussion instead of reporting the experimental facts and then base some discussion on them.
- Large parts of the results are then discussed as if these were cooling experiments, which is misleading. It would have been very, very helpful if Figures like 2a and c (btw: the a,b,c is missing from the actual figure and only available in the caption) would first have been used to explain how the signals are interpreted along the timeline of the experiment. I would assume that upon freezing, conceptually, Ice Ih + solute-bearing solid would crystallize until pressure is high enough that denser Ice II (or III) crystallization would be induced to buffer the expansion due to Ice II formation. This would be the pressure on the left of Fig 2a,c at which the heating experiments then commence and so, the near-horizontal curve parts in Fig 2 are actually representing heating the 3-solids assemblage before the formation of liquid upon heating (not last freezing of it is implied in line 123). Stating this concept in line with the actual experimental protocol would help the reader understand the reasoning and interpretation much better.
- In fact, the data as such do not tell if the Ice II (III) / Ih boundary was reached, the authors should provide arguments for that (e.g., coincidence with previous T-P data on this phase transition). Otherwise, similar heating curves would be obtained for Ice Ih + solute-bearing solid systems, right?
- In that sense, large parts of the (partly redundant/circular) discussion could be shortened if it would be made crystal-clear in the beginning that the “cenotectic” always is the intersection of the Ice II (or III) / Ih phase boundary with the cotectic (which is the proper IUPAC term rather than “eutectic line”). Then, large parts of the discussion would almost be obsolete and boil down to providing arguments that this boundary was indeed encountered in the experiment. Furthermore, it should be stated earlier on (and not be somewhat hidden late in the discussion) that the experiments do not provide any evidence on the T-P trajectory of next cotectic at pressures beyond the “cenotectic”, i.e., if the conceptual idea of Fig. 1 that it'll have a different T-P dependence is correct (btw. arguing with a single-component Clapeyron slope for a two-component, two solids + liquid system is not very defensible if the assumptions are not spelled out).
- Also, the discussion of the “22K” offset could be done much more rigorously. As the boundaries II/Ih and III/Ih have slopes

in P-T, the intersection of the lh/L curve, shifted by the eutectic temperature at 0.1 MPa and intersected with the respective ice phase boundaries (e.g., taking the parameterizations of Dunaeva et al., 2010; Solar System Research, 44, 202-222), would be much more useful than the simplistic approach of Fig. 2b. Within the uncertainty of the Dunaeva fits, the agreement is remarkably good.

- Regarding the latter, in addition: indeed, the Ice-lh melting behavior seems to control the P-T trajectory of the cotectic (no real surprise as it is associated with the largest volume change). However, the shape of the solute-bearing solid's liquidus also plays a role in modulating that broad behavior, so the 22K are only approximate (in spite of the "remarkably good" when using Dunaeva) and may not be an as general rule as inferred from the rather small data base.
- Besides the reader-unfriendly text- and argument-organization, often, the wording is not very accurate (eutectic line -> cotectic, "granularity" (line 144) -> resolution, and so on).
- The parts on the findings with NaCl etc. that may indicate new hydrate phases dilute the overall results and discussion. These could easily go into the supporting material and just briefly be referenced in the main text. I also speculate that parts of the strange curve shapes may rather be the results of the presence of "normal" hydrate phases. The melting curve's T-P trajectory would be influenced by the fact that starting with a 1bar near-eutectic composition would move the composition towards the solute-rich side of the eutectic in the case of NaCl at higher P (taking the Chang et al. results from the same group who reported a shift to lower NaCl-contents on the cotectic with increasing P), i.e., the melting behavior (which solid starts to melt first) will be different from starting on the water-rich side. Again, a thorough discussion of the behavior along the experimental time coordinate would be very helpful.
- The section "Implications for materials thermodynamics and generalized definition of the cenotectic" is not adding a lot of value and can be left out.
- In the section on "Implications for planetary science", the reasoning and assumptions behind the structures displayed in Fig. 3 and their link to the experimental results are not really apparent and the authors should guide the reader. In that sense, in its current form, the text does not provide new insights based on the new experimental results.

Reviewer #4

(Remarks to the Author)

In this work, the authors experimentally explored the effect of pressure on the eutectic in aqueous systems. According to the authors' findings, as the pressure increases, the eutectic point is transformed into an eutectic line, which ends in a "cenotectic" point, which is characterized by the cenotectic temperature and cenotectic pressure and depends on the solute type. The relevance of this work is that the discovery of the cenotectic point plays a central role in understanding complex multiphase equilibria and denotes the limits at which a liquid phase is stable invariantly. From a thermodynamic viewpoint, this point represents a new thermodynamical coordinate on the Clapeyron diagram for mixtures and also extends the application of thermodynamics to other sceneries.

The reported information (i.e., experimental details, data interpretation, etc.) is broadly explained and discussed, providing the information needed to reproduce the experiments.

Some minor details to be considered to improve this work are:

1. It suggests fitting the curves reported in Fig. 2 by using the glass transition temperature hyperbola proposed by Patrone et al. (P.N. Patrone, A. Dienstfrey, A.R. Browning, S. Tucker, S. Christensen, Uncertainty quantification in molecular dynamics studies of the glass transition temperature, Polymer 87 (2016) 246–259.) rather than polynomial functions. The advantage of the glass transition temperature hyperbola is that it provides more accurate transition coordinates.
2. In the text, the authors use T_c and P_c as the cenotectic temperature and pressure and z_c as the cenotectic depth. It is advised to replace them with other symbols because it causes confusion with fluids' critical temperature, pressure, and compressibility.
3. It is important to discuss the effect of the initial concentration used in the experiments.
4. According to the experimental results, the authors claim, " We observe that when no solute-bearing high-pressure phases exist, the cenotectic occurs consistently 22 ± 2 K below the 1 bar eutectic and at approximately 213 ± 3 MPa" This statement is valid for the explored systems but the authors need to provide more evidence to be generalized.

Reviewer #5

(Remarks to the Author)

The authors have submitted a manuscript where they considered the liquid limit of aqueous salt solutions at high pressures. The manuscript started with the measurement of experimental data followed by a discussion thereof and ended with implications of the measurements to various different fields.

The manuscript has been submitted to Nature Communications and therefore the intended audience is broader than that of a traditional thermodynamics/thermophysical journal. The authors tried to but at times struggled to present information to the wide audience but still show the scientific merits of the manuscript. The authors need to consider the main contribution of the manuscript and ensure that this comes through strongly. At times the manuscript is lacking in focus and therefore the impact

is difficult to judge. The experimental measurements are not very well presented (see below) yet some of the discussions are at times superficial. Further, a significant amount of text is dedicated to the applications but I am not convinced these findings are a significant scientific contribution (I may be wrong as I am not an expert in many of the applications e.g. planetary science and cryopreservation).

The manuscript has potential but, in my opinion, needs some improvement before it can be published.

General comments:

Please define terms and abbreviations used at first use.

What is the mathematical definition of the cenotectic point? Similar to the critical point of a pure component where $\partial^2 P / (\partial v^2) = 0$ and $\partial P / \partial v = 0$ at $T = T_c$, how does one define the cenotectic point?

The thermodynamic basis is poorly explained. At times the Gibbs Phase Rule is mentioned but the underlying thermodynamics could have been better presented.

How do the results presented in Table 1 relate to known thermodynamic behaviour of the various systems? The authors allude to this in their discussion but a clearer explanation is needed, in my opinion.

Method used for measurements and presentation of experimental results is not fully clear and therefore there are some concerns regarding the credibility of the results.

The authors state that this is the first time these types of measurements have been conducted yet the experimental method essentially only referred to in the main manuscript and not described in detail in the supplementary information. I would recommend a more detailed exposition of the experimental method to illustrate the approach used, especially as part of the contribution of the manuscript is the 'first-of-their-kind' measurements.

Especially considering the audience, the authors need to expand how their constant volume, constant bulk composition measurements will give an accurate description of the cenotectic point. Further, how does the accuracy of the composition loaded influence the measurements? The authors could project their measurement isochore on Figure 1. Part of this information can be provided in the supplementary information.

From the experimental method how can the authors prove that the true cenotectic point is obtained?

The authors are requested to do a full uncertainty analysis as outlined by the "Guide to the expression of uncertainty in measurement" produced by Working Group 1 of the Joint Committee for Guides in Metrology (JCGM/WG 1). The uncertainty in the temperature, pressure and composition should be provided taking into account all factors influencing the results. It is standard practice for thermodynamic data to include these measurements. The standard deviation provided here are, according to my understanding, the standard deviations from the three repeat runs and do not constitute to experimental uncertainties.

Version 1:

Reviewer comments:

Reviewer #2

(Remarks to the Author)

I thank the authors for thoroughly addressing my comments and recommend that the manuscript be accepted for publication in its current form.

Reviewer #3

(Remarks to the Author)

The revision has clarified many things that I criticized in the earlier version; namely, the much clearer documentation of experimental procedures and interpretation of the experimental curves now allows readers to form an own opinion on the results. All further critique that I could raise now (e.g., regarding style and clarity of writing as well as clarity of scientific details, such as why a thicker ice crust would enhance fluid circulation and how much) would be rather subjective / express my personal preferences, so I refrain from suggesting these points for minor revisions. In my opinion the manuscript can be accepted now.

Reviewer #4

(Remarks to the Author)

In this revised version, the authors carried out all the questions and queries

Reviewer #5

(Remarks to the Author)

The authors have submitted an improved manuscript as well as a detailed description of the changes made to address the concerns of the reviewers. The resubmitted manuscript is an improvement of the previous submission but in my opinion the manuscript still requires improvement before it can be considered for publication.

1. The authors need to be clear in the manuscript that they are considering aqueous salt solutions. Throughout they talk about liquids but the liquid considered is water. This needs to be clearer in the text. Also, throughout the authors need to refer to the liquid as water and be specific when they refer to the salt e.g. line 191 'it' is indeed the salt.

2. The authors have attempted to improve the clarity of the manuscript by including the methodology. This is however given at the end of the text with no reference to it in the results section that starts immediately after the introduction. One of the main contributions of the manuscript is the establishment of a new method. Therefore, I strongly feel that the methodology section

should be before the results; the reader needs to know how the experiments were conducted before the outcomes of the experiments are discussed.

3. The cooling and subsequent heating profiles have been given in the experimental section. How do the authors know that they are at equilibrium, especially at the start of the experiment. If the mixture is cooled down for only 45 min, is this enough to get to equilibrium? Personal experience suggests that equilibrium, especially those including a phase change where there is no mixing can take very long to attain, hours, sometimes even days, not just minutes. Figures S3 and S5 do not start at the same pressure when heated; is this not an indication that the experiments were not at equilibrium? I am not convinced regarding the results for the salts in Figure 2(c).

Some minor comments

4. Please add page numbers and give figures and tables as soon as possible after first mention.

5. The figure captions are very long. Be careful of including descriptions and interpretations in the captions.

Version 2:

Reviewer comments:

Reviewer #3

(Remarks to the Author)

Please see attached pdf

Version 3:

Reviewer comments:

Reviewer #3

(Remarks to the Author)

Thanks for the positive feedback on my previous comments, and I'd be happy to be acknowledged.

Obviously the revised figure fits the data and text doesn't contradict the new Figures, but: I wonder if you don't give away an opportunity to show the (claimed) generality and to make even some predictions. Sorry for bringing that up as the quick response seemed to indicate some pressure to get the paper out. Yet, I think for a "high impact" journal also the content should make as much out of the content as possible with the insights obtained during the study. And, I like the results and would find it a pity if you let the chance pass to make this more "seminal".

So, my previous suggestion with three surfaces on Fig. 1 showed that the experimental curves could align differently (quite significantly, actually, your Figure 2) with the ice diagram with the mole fraction of solute particles. Assuming full dissociation of electrolytes and none of urea, these are MgSO_4 0.03, Na_2SO_4 0.016, Na_2CO 30.03 and KCl 0.11, urea 0.127, if I didn't miscalculate that. One sees easily, that these numbers align nicely with the positions on Fig 2a and I think there is quite some benefit from adding this insight to Fig 1.

One can now speculatively predict that there may be cases (with high eutectic molar fractions of solute particles) for which one would predict a cenotectic with ice II stability (your original Fig. 1) and according possible variations in cenotectic pressure etc.

I attach a slightly changed suggestion for what could be modified in Fig 1 to highlight this (and it needs a disclaimer that it is schematic; yet, all salts except MgCl_2 seem to follow a +/- similar eutectic curve at 0.1 MPa if plotted in mole fraction space and assuming full dissociation, so it seems very acceptable to stick to the suggested design). Please consider this seriously as a starting point for a revision of Fig. 1 given the points above (and a few minor explanations added to the text and maybe in the results).

Of course, it is your paper; if you feel that the above thoughts are misleading or likely/possibly "too wrong" to be considered, then please reject the suggestion but with a good explanation why.

RESPONSE TO REVIEWERS

We sincerely thank all five reviewers for their diligent evaluations. We have responded to all of their comments and made substantial corresponding revisions to both the main text and the supplementary information. As a result, we believe the manuscript has been significantly improved, and we hope that the reviewers and editor will now find it suitable for publication.

In the text below, we reproduce each of the original reviews in black, and provide point-by-point responses in blue. Given the extensiveness of the edits, in the marked-up revised manuscript we have highlighted the areas of most significant revision. We thank the reviewers again for their time and consideration.

Reviewer #1 (Remarks to the Author):

The manuscript “On the equilibrium limit of liquid stability in pressurized aqueous systems” presents the results of an experimental study, involving the isochoric freezing of binary aqueous brines, which demonstrates the existence of a “cenotectic” defined as the “invariant point occurring at the lowest temperature at which the liquid phase, for any value of concentration, pressure, or other thermodynamic forces acting on the system, remains in stable equilibrium”. Although this work was limited to binary aqueous brines, the manuscript highlights that the results can be extended more generally to other systems, where the location of the cenotectic will be governed by intersection of phase equilibrium lines with opposite Clapeyron slopes. The existence of a cenotectic is demonstrated to be significant to icy ocean worlds since it governs the depth at which the final remnant of a liquid ocean solidifies. Implications for cryopreservation are also highlighted.

The manuscript represents an important contribution to the scientific literature and introduces a key concept that has implications across a diverse range of fields. The results are compelling, and the methodology is well-founded. However, there are a few areas where the manuscript could be improved.

We thank the reviewer for their close reading of the manuscript. We are pleased to hear they find the results compelling, and are very happy to incorporate their suggested improvements.

Specific recommendations for minor edits are included in the attachment, but more significant comments are summarized here. We recommend the manuscript be published after these comments are sufficiently addressed.

-Figure 1:

- Please label the concentration axis and include the appropriate limits. Currently the axis value is 0 at both ends (corresponding to the 0 value for temperature and pressure).
- Since the variables P-T-x are used throughout the manuscript, I recommend including them in the label, Example: Pressure, P (GPa)

- Please define m_e , m_c , P_c in the figure caption

We have executed all the changes recommended above for Figure 1, thank you for the suggestions!

- Consider including a 2D cutout (or multiple 2D cutouts) of the figure intersecting at point C

We thank the reviewer for this suggestion, and we heavily considered the potential merit of 2D slices. However, we concluded that they may only introduce undue confusion— this is because the cenotectic point of interest is a four-phase equilibrium invariant, and a 2D slice can only effectively convey up to three-phase equilibrium. In light of this, we elected to maintain only the 3D visualization.

- It's not clear that m_e is different from m_c given your experiments, please elaborate on this point somewhere in the manuscript

We thank the Reviewer for identifying this oversight, and we have added a brief paragraph explaining that, starting at the eutectic concentration m_e , we expect based on previous work that the concentration of the liquid phase to continually decrease with increasing pressure until m_k (the cenotectic concentration as now marked) is reached.

-Discontinuity:

- The term discontinuity is used to describe the conditions at which the cenotectic is reached. This is not formally a discontinuity in the eutectic curve itself but perhaps a “kink” or more explicitly a “discontinuity in the slope”?

We thank the reviewer for this comment, and have updated the language to employ “discontinuity in the slope” throughout.

-Table 1:

- Consider emphasizing that the initial concentration is the eutectic concentration at 1 atm, since it's not immediately apparent from the table.
- For clarity, please write out "Difference between Eutectic Temperature at 0.1 MPa and cenotectic temperature" instead of or in addition to ΔT

We thank the reviewer for these Table comments, and have incorporated all of them.

-Figure 2:

- Please label plots with a, b, c

- In addition to Delta T, please write out "Difference between Eutectic and Cenotectic Temperature"
- Please include variables along with axis labels, Example: Temperature, T (K)

We thank the reviewer for these figure comments, and have incorporated all of them.

-Table 2:

- The gravitational accelerations and therefore depth of the cenotectic for Titan and Europa appear to be switched

Thanks for this good catch, the table has been corrected.

- Consider including the source for the gravitational acceleration, particularly for the exoplanets

The gravitational accelerations are provided in the reference cited in the text (Ojha et al. 2022, Nat. Comms). We added it in the caption as well for clarity. Thanks.

-Figure 4:

- The gravitational accelerations and therefore depth of the cenotectic for Titan and Europa appear to be switched

Thanks again, the figure has been corrected.

-Terminology

One recurring but minor issue throughout the manuscript is the lack of consistency and precision in terminology:

- “eutectic line” vs. “eutectic curve” vs. “P-T curve” vs. “P-T trajectory” vs. “eutectic pressure-temperature profile”. To minimize confusion, I recommend adopting a single term for the “eutectic curve” and using it throughout the manuscript.

We unified the description to “eutectic curve” throughout the manuscript for clarity. Thanks for the suggestion.

- If using the term “P-T coordinates” specify if these are “P-T coordinates of the eutectic points” or “P-T coordinates of the cenotectic point”.

Thank you, we have done so.

- Similarly for atmospheric pressure 1 atm, 1 bar, 0.1 MPa?

Thank you, we have unified the description of atmospheric pressure as 0.1 MPa.

- T-P vs P-T (I recommend P-T)

Thank you, we now use P-T throughout.

- P-T-X vs. P-T-x

Thank you, we now use P-T-x throughout.

Reviewer #2 are duplicates of Reviewer #1. We thank both reviewers again for their consideration and suggestions.

Reviewer #3 (Remarks to the Author):

Zarriz et al. report first-of-their-kind experimental data on low-temperature (T) / high pressure (P) phase relations of aqueous salt systems, namely between the different ice polymorphs and salt-bearing solids. They claim to determine the T-P coordinates of the low-temperature endpoint of the cotectic between ice and a solute-bearing solid, which they name the “cenotectic” and suggest that this is the lowest possible temperature at which the respective aqueous salt solution can stably exist. They then explore implications for thermodynamics of such systems and applications to “icy moon” environments.

The experimental findings as such are very interesting and novel. Yet, I had a very hard time preparing this review as the organization of the manuscript is poor. In my opinion it requires a major re-write / thorough major revisions, followed by another round of reviewing, before it can be considered for publication in a leading peer-reviewed journal. Namely, a clear documentation of the experimental approach and its likely first-order interpretation is largely absent although it would be absolutely essential to make the text more readable and the work reproducible. Regarding discussion of the implications a much stronger and clearer connection to the experimental results should be made.

We are very grateful for the in-depth review provided, and we have reorganized various aspects of the manuscript to enhance clarity, with particular regard to the method employed. We hope that the Reviewer now finds the manuscript easier to digest, and thank them again for their close reading of the work.

Some of my concerns in detail are:

- Perhaps my biggest criticism is that the authors do not clearly (and not even very briefly) document how they did the experiments and that the remaining descriptions and wording in the text are really misleading in this respect. If I understood it correctly – and it is really unacceptable that one has to read a previous publication (Chang et al.) to get to this point – the experiments are actually NOT freezing experiments but rather melting / heating experiments at very low temperatures. The experimental charge is first frozen and then slowly heated and phase transitions are recorded as changes of temperature/pressure slopes upon heating in the isochoric system. Unfortunately, this is never clearly spelled out (a “Methods” or “Experimental” section would have helped). In fact, much of what should be part of such an early-on description of what was done and observed comes as interpretation in the discussion instead of reporting the experimental facts and then base some discussion on them.
- Large parts of the results are then discussed as if these were cooling experiments, which is misleading. It would have been very, very helpful if Figures like 2a and c (btw: the a,b,c is missing from the actual figure and only available in the caption) would first have been used to explain how the signals are interpreted along the timeline of the experiment. I would assume that upon freezing, conceptually, Ice Ih + solute-bearing solid would crystallize until pressure is high enough that denser Ice II (or III) crystallization would be induced to buffer the expansion due to Ice II

formation. This would be the pressure on the left of Fig 2a,c at which the heating experiments then commence and so, the near-horizontal curve parts in Fig 2 are actually representing heating the 3-solids assemblage before the formation of liquid upon heating (not last freezing of it is implied in line 123). Stating this concept in line with the actual experimental protocol would help the reader understand the reasoning and interpretation much better.

The following answer addresses the last 2 points:

We agree with the reviewer entirely that the experimental details were not sufficiently documented and that the way it was previously described could have been left unclear that these were isochoric freezing AND warming/melting experiments, in which final equilibrium data is recorded during the melting process.

We have addressed these important concerns by rephrasing the entire manuscript accordingly; clarifying aspects of the methods section; providing much more comprehensive example time-series data of the entire freezing-melting process in the Supplementary Information; and adding a comprehensive uncertainty analysis in the Supplementary Information.

We thank the reviewer for flagging this– the manuscript has been substantially improved as a result.

- In fact, the data as such do not tell if the Ice II (III) / I_h boundary was reached, the authors should provide arguments for that (e.g., coincidence with previous T-P data on this phase transition). Otherwise, similar heating curves would be obtained for Ice I_h + solute-bearing solid systems, right?

We thank the reviewer for this comment. The cenotectic points measured in this work always occur around 210MPa, which corresponds to the ice I_h-II and I_h-III solid-solid phase transition. It is reasonable to believe that ice II and III grow pure, as ice I_h and ice VI partition coefficients for salts have been measured to be below 5E-3 (Journaux et al. 2017, EPSL 463, 36–47). Therefore it is reasonable to explain the location of the cenotectic transition to be along the quasi-isobaric I_h-III or I_h-II transitions around 209 MPa (Journaux et al. 2020, Journal of Geophysical Research: Planets 125 (1), e2019JE006176). We added discussion to the manuscript to clarify this point.

Additionally, to aid interpretation, we have plotted the pure-water ice I_h - ice II and ice I_h - ice III phase boundaries in Figures 2a and 2b. We thank the reviewer again for this comment.

- In that sense, large parts of the (partly redundant/circular) discussion could be shortened if it would be made crystal-clear in the beginning that the “cenotectic” always is the intersection of the Ice II (or III) / I_h phase boundary with the cotectic (which is the proper IUPAC term rather than “eutectic line”). Then, large parts of the discussion would almost be obsolete and boil down to providing arguments that this boundary was indeed encountered in the experiment. Furthermore, it should be stated earlier on (and not be somewhat hidden late in the discussion) that the experiments do not provide any evidence on the T-P trajectory of next cotectic at pressures

beyond the “cenotectic”, i.e., if the conceptual idea of Fig. 1 that it’ll have a different T-P dependence is correct (btw. arguing with a single-component Clapeyron slope for a two-component, two solids + liquid system is not very defensible if the assumptions are not spelled out).

We thank the reviewer for these comments. We have added a line in the beginning of the manuscript clarifying that, **for the systems studied here**, the cenotectic will occur at the intersection of the ice Ih / ice II (or III) phase boundary and the eutectic curve / cotectic. We have also significantly expanded our discussion of the data in Figure 2 in general to reflect these points.

- Also, the discussion of the “22K” offset could be done much more rigorously. As the boundaries II/Ih and III/Ih have slopes in P-T, the intersection of the Ih/L curve, shifted by the eutectic temperature at 0.1 MPa and intersected with the respective Ice phase boundaries (e.g., taking the parameterizations of Dunaeva et al., 2010; Solar System Research, 44, 202-222), would be much more useful than the simplistic approach of Fig. 2b. Within the uncertainty of the Dunaeva fits, the agreement is remarkably good.

We agree with the reviewer, and for clarity have added the relevant boundaries of the pure-water phase diagram to Figures 2a and 2b. We used the SeaFreeze derived phase diagram (Journaux et al. 2020, Journal of Geophysical Research: Planets 125 (1), e2019JE006176), as it combines the self-consistent Gibbs energy descriptions for ice II and III from recent XRD data with the Ih Gibbs description from Feistel and Wagner (2006), which all reproduce well experimental data for the Ih-II and Ih-III phase transitions. We also rewrote the related part of the discussion accordingly. Thanks very much for the comment.

- Regarding the latter, in addition: indeed, the Ice-Ih melting behavior seems to control the P-T trajectory of the cotectic (no real surprise as it is associated with the largest volume change). However, the shape of the solute-bearing solid’s liquidus also plays a role in modulating that broad behavior, so the 22K are only approximate (in spite of the “remarkably good” when using Dunaeva) and may not be an as general rule as inferred from the rather small data base.

We thank the reviewer for this comment, and have added additional discussion to the end of this section stating more clearly that this 22K offset is simply a rule of thumb, with applicability only to systems for which the ice Ih phase boundary is much more pressure sensitive than the solute-bearing solid phase boundaries.

- Besides the reader-unfriendly text- and argument-organization, often, the wording is not very accurate (eutectic line -> cotectic, “granularity” (line 144) -> resolution, and so on).

We thank the reviewer for calling these out. We have corrected “granularity” to be “resolution”. At the advice of other reviewers, we have unified our language as “eutectic curve”, as opposed to “eutectic line”, “eutectic trajectory”, etc. We now also mention in the introduction that here the eutectic curve is equivalent to a cotectic.

- The parts on the findings with NaCl etc. that may indicate new hydrate phases dilute the overall results and discussion. These could easily go into the supporting material and just briefly be referenced in the main text. I also speculate that parts of the strange curve shapes may rather be the results of the presence of “normal” hydrate phases. The melting curve’s T-P trajectory would be influenced by the fact that starting with a 1bar near-eutectic composition would move the composition towards the solute-rich side of the eutectic in the case of NaCl at higher P (taking the Chang et al. results from the same group who reported a shift to lower NaCl-contents on the cotectic with increasing P), i.e., the melting behavior (which solid starts to melt first) will be different from starting on the water-rich side. Again, a thorough discussion of the behavior along the experimental time coordinate would be very helpful.

We thank the reviewer for this comment, but must very respectfully disagree with them on the value of the NaCl findings. We believe the discussion of possible new hydrates to be integral to a comprehensive understanding of the cenotectic concept, and believe further that the distinct differences between the P-T curves exhibiting potential intermediate hydrates and the simpler curves shown in Figure 2a highlight the diversity of potential phase behaviors that future practitioners of isochoric freezing and melting may encounter. We have tried to enhance the level of discussion in this section in order to increase clarity.

- The section “Implications for materials thermodynamics and generalized definition of the cenotectic” is not adding a lot of value and can be left out.

We thank the Reviewer for this comment, but again feel that this section broadens the impact of the results and also provides the definition/logic necessary for the cenotectic concept to be rigorously understood and applied in the future. In fact, this section in particular was requested by previous reviewers, who argued to us that without a rigorous discussion of the possible manifestations of the cenotectic, the risk that it may be misinterpreted and misapplied in the future is much higher. In order to supplement the rigor of this section however, we have added an additional Supplementary Note (#5) further clarifying formal aspects of the cenotectic within the context of Gibbs’ Phase Rule.

- In the section on “Implications for planetary science”, the reasoning and assumptions behind the structures displayed in Fig. 3 and their link to the experimental results are not really apparent and

the authors should guide the reader. In that sense, in its current form, the text does not provide new insights based on the new experimental results.

We thank the reader for flagging this, and have rephrased the planetary science section to better guide the reader. We sincerely hope that reviewer will now find it both clearer and more compelling, as we believe these (simple!) applications of the cenotectic concept to planetary science to be very impactful.

Reviewer #4 (Remarks to the Author):

In this work, the authors experimentally explored the effect of pressure on the eutectic in aqueous systems. According to the authors' findings, as the pressure increases, the eutectic point is transformed into an eutectic line, which ends in a "cenotectic" point, which is characterized by the cenotectic temperature and cenotectic pressure and depends on the solute type. The relevance of this work is that the discovery of the cenotectic point plays a central role in understanding complex multiphase equilibria and denotes the limits at which a liquid phase is stable invariantly. From a thermodynamic viewpoint, this point represents a new thermodynamical coordinate on the Clapeyron diagram for mixtures and also extends the application of thermodynamics to other sceneries.

The reported information (i.e., experimental details, data interpretation, etc.) is broadly explained and discussed, providing the information needed to reproduce the experiments.

Some minor details to be considered to improve this work are:

1. It suggests fitting the curves reported in Fig. 2 by using the glass transition temperature hyperbola proposed by Patrone et al. (P.N. Patrone, A. Dienstfrey, A.R. Browning, S. Tucker, S. Christensen, Uncertainty quantification in molecular dynamics studies of the glass transition temperature, *Polymer* 87 (2016) 246–259.) rather than polynomial functions. The advantage of the glass transition temperature hyperbola is that it provides more accurate transition coordinates.

We thank the reviewer for this comment and their interesting suggestion. After reading the suggested work, we respectfully conclude that the method described in there would not quite be justifiable for our application here, because that method targets a smooth and continuous second-order transition (the glass transition), whereas we target here the discrete intersection of liquidus or cotectic curves (both of which mark first-order transitions). However, we agree with the reviewer that more robust uncertainty quantification is needed, as was also suggested by Reviewer 5 below. As such, we have added a comprehensive new uncertainty analysis to the SI, according to the "Guide to the expression of uncertainty in measurement", produced by Working Group 1 of the Joint Committee for Guides in Metrology (JCGM/WG 1). We have accordingly updated the uncertainty values in all of the tables where temperature-pressure coordinates are reported. We thank the reviewer again for highlighting the need for a more robust uncertainty analysis.

2. In the text, the authors use T_c and P_c as the cenotectic temperature and pressure and z_c as the cenotectic depth. It is advised to replace them with other symbols because it causes confusion with fluids' critical temperature, pressure, and compressibility.

We thank the reviewer for this comment. We agree that this might cause confusion and have changed them to T_K , P_K , etc., to avoid confusion with the critical properties.

3. It is important to discuss the effect of the initial concentration used in the experiments.

We thank the reviewer for this note. We have enhanced the discussion of our choice to use 0.1 MPa eutectic concentration as our initial concentration, and further clarified the chosen concentration in Table 1. We have also added commentary in Supplementary Note 1 on the relative insensitivity of the eutectic curve's T-P trajectory to experimental uncertainty in starting concentration.

4. According to the experimental results, the authors claim, "We observe that when no solute-bearing high-pressure phases exist, the cenotectic occurs consistently 22 ± 2 K below the 1 bar eutectic and at approximately 213 ± 3 MPa" This statement is valid for the explored systems but the authors need to provide more evidence to be generalized.

We thank the reviewer for this comment, and have added additional discussion to the end of this section stating more clearly that this 22K offset is simply a rule of thumb, with applicability only to systems for which the ice I_h phase boundary is much more pressure sensitive than the solute-bearing solid phase boundaries.

Reviewer #5 (Remarks to the Author):

The authors have submitted a manuscript where they considered the liquid limit of aqueous salt solutions at high pressures. The manuscript started with the measurement of experimental data followed by a discussion thereof and ended with implications of the measurements to various different fields.

The manuscript has been submitted to Nature Communications and therefore the intended audience is broader than that of a traditional thermodynamics/thermophysical journal. The authors tried to but at times struggled to present information to the wide audience but still show the scientific merits of the manuscript. The authors need to consider the main contribution of the manuscript and ensure that this comes through strongly. At times the manuscript is lacking in focus and therefore the impact is difficult to judge. The experimental measurements are not very well presented (see below) yet some of the discussions are at times superficial. Further, a significant amount of text is dedicated to the applications but I am not convinced these findings are a significant scientific contribution (I may be wrong as I am not an expert in many of the applications e.g. planetary science and cryopreservation).

The manuscript has potential but, in my opinion, needs some improvement before it can be published.

We thank the reviewer for their time and consideration here – we are very happy to incorporate the suggested improvements in order to bring the manuscript to Nature Communications quality. Regarding text spent discussing other applications, in order to enhance narrative clarity, at the suggestion of the other reviewers and the editor, we have expanded our discussion of planetary science applications to better guide the reader, and removed the brief section on cryopreservation.

General comments:

- Please define terms and abbreviations used at first use.

Thank you, we have done so.

- What is the mathematical definition of the cenotectic point? Similar to the critical point of a pure component where $\left. \frac{\partial^2 P}{\partial v^2} \right|_{(T=T_c)} = 0$ and $\left. \frac{\partial P}{\partial v} \right|_{(T=T_c)} = 0$, how does one define the cenotectic point?

The cenotectic does not lend itself to straightforward mathematical definition in the same way that the critical point does, because at the critical point, only one phase exists, whereas at the cenotectic point, depending on the composition of the solution and the modes of thermodynamic work affecting it, many phases co-exist.

We have thus in the manuscript described it phenomenologically, rather than mathematically. We note that this too is how the eutectic is typically defined, for the same reason.

However, in an effort to clarify formal aspects of the cenotectic definition, we have added a new Supplementary Note 5 that gives a more rigorous technical definition within the context of Gibbs' Phase Rule. We anticipate that between the extensive discussion and definition in the manuscript and this new Supplementary Note, the definition of the cenotectic should now be clear to the reader. We thank the reviewer for this note.

- The thermodynamic basis is poorly explained. At times the Gibbs Phase Rule is mentioned but the underlying thermodynamics could have been better presented.

We thank the reviewer for this comment, and have worked to clarify the underlying thermodynamic basis. As mentioned in the previous response, we have also included Supplementary Note 5 further clarifying the relationship between the Phase Rule and cenotectic.

- How do the results presented in Table 1 relate to known thermodynamic behaviour of the various systems? The authors allude to this in their discussion but a clearer explanation is needed, in my opinion.

We thank the reviewer for this comment, and agree that the heading labeling in Table 1 was insufficiently clear. We have amended this for clarity. We now list the initial brine concentration in the experimental system, which is the known eutectic concentration at 0.1 MPa; the according eutectic temperature at 0.1 MPa; our measured cenotectic temperatures and pressures; and the difference in temperature between our measured cenotectic temperature and the known 0.1 MPa eutectic temperature. Thank you again.

- Method used for measurements and presentation of experimental results is not fully clear and therefore there are some concerns regarding the credibility of the results.
- The authors state that this is the first time these types of measurements have been conducted yet the experimental method essentially only referred to in the main manuscript and not described in detail in the supplementary information. I would recommend a more detailed exposition of the experimental method to illustrate the approach used, especially as part of the contribution of the manuscript is the 'first-of-their-kind' measurements.

We agree with these last two comments, and have added various additional methodological details throughout the manuscript (in the main text, the methods, and the Supplementary Information alike) that provide more in-depth description of the experimental techniques and analysis. We also provide a new supplementary note dedicated to uncertainty analysis, and we believe this information in sum should allow other groups to fully reproduce the experiments. We thank the reviewer for flagging this.

- Especially considering the audience, the authors need to expand how their constant volume, constant bulk composition measurements will give an accurate description of the cenotectic point. Further, how does the accuracy of the composition loaded influence the measurements? The authors could project their measurement isochore on Figure 1. Part of this information can be provided in the supplementary information.

We thank the reviewer for this comment, and have worked throughout the main text and Supplementary Information to clarify how and why the isochoric freezing and melting method leads us to the cenotectic point. We believe that with better description of the methods/approach throughout, and new details in Supplementary Notes 1, 2 and 5, it should now be sufficiently clear. We have also explicitly addressed in Supplementary Note 1 that experimental uncertainty in concentration can generally affect the *phase fractions of the phases involved in the eutectic assemblage*, but not the P-T trajectory of the equilibrium eutectic curve, because the system has only one intensive thermodynamic degree of freedom per Gibbs' Phase Rule.

We have furthermore highlighted in the Figure 1 caption that the P-T trajectory of the measured isochore is given by the thick magenta eutectic curve between points (e) and (□) in the Figure 1 phase diagram.

- From the experimental method how can the authors prove that the true cenotectic point is obtained?

As mentioned in the text, our confidence that the measured invariant points truly represent the cenotectic is ultimately limited by our understanding of the phase behavior of the system as a whole. If indeed the isochoric trajectory is originating from the lowest-temperature eutectic point that exists at atmospheric pressure, based on our thermodynamic arguments, we reason that our confidence in interpreting the cenotectic correctly is high. However, if, for a given system, a different atmospheric-pressure eutectic was to be discovered in the future at *lower* temperatures, starting the isochoric trajectory from *that* eutectic concentration would lead to the cenotectic. As such, we operate here with the best information available, but acknowledge that (as in most aspects of materials science and discovery), future research may either support, update, or change our conclusions.

- The authors are requested to do a full uncertainty analysis as outlined by the "Guide to the expression of uncertainty in measurement" produced by Working Group 1 of the Joint Committee for Guides in Metrology (JCGM/WG 1). The uncertainty in the temperature, pressure and composition should be provided taking into account all factors influencing the results. It is standard practice for thermodynamic data to include these measurements. The standard deviation provided here are, according to my understanding, the standard deviations from the three repeat runs and do not constitute to experimental uncertainties.

We thank the reviewer for this comment, and agree with the need for more extensive uncertainty analysis. We now include an extensive uncertainty analysis in the SI, following the JCGM guidelines, and we have updated all tables reporting temperature-pressure data accordingly. Thanks again for the suggestion.

RESPONSE TO REVIEWERS

We sincerely thank all five reviewers for re-evaluating our manuscript. We have responded to all of their comments and made substantial corresponding revisions to both the main text and the supplementary information. As a result, we believe the manuscript has been significantly improved, and we hope that the reviewers and editor will now find it suitable for publication.

In the text below, we reproduce each of the original reviews in black, and provide point-by-point responses in blue. Given the extensiveness of the edits, in the marked-up revised manuscript we have highlighted the areas of most significant revision. We thank the reviewers again for their time and consideration.

Reviewer #2 (Remarks to the Author):

I thank the authors for thoroughly addressing my comments and recommend that the manuscript be accepted for publication in its current form.

We thank the Reviewer for their time and consideration, and agree with their assessment.

Reviewer #3 (Remarks to the Author):

The revision has clarified many things that I criticized in the earlier version; namely, the much clearer documentation of experimental procedures and interpretation of the experimental curves now allows readers for form an own opinion on the results. All further critique that I could raise now (e.g., regarding style and clarity of writing as well as clarity of scientific details, such as why a thicker ice crust would enhance fluid circulation and how much) would be rather subjective / express my personal preferences, so I refrain from suggesting these points for minor revisions. In my opinion the manuscript can be accepted now.

We thank the Reviewer for their time and consideration, and agree with their assessment.

Reviewer #4 (Remarks to the Author):

In this revised version, the authors carried out all the questions and queries

We thank the Reviewer for their time and consideration, and agree with their assessment.

Reviewer #5 (Remarks to the Author):

The authors have submitted an improved manuscript as well as a detailed description of the changes made to address the concerns of the reviewers. The resubmitted manuscript is an improvement of the previous submission but in my opinion the manuscript still requires improvement before it can be considered for publication.

We thank the Reviewer for their time and consideration here, and are happy to make further modifications in response to their new comments. Please see below.

1. The authors need to be clear in the manuscript that they are consider aqueous salt solutions. Throughout they talk about liquids but the liquid considered is water. This needs to be clearer in the text. Also, throughout the authors need to refer to the liquid as water and be specific when they refer to the salt e.g. line 191 'it' is indeed a the salt.

We thank the reviewer for this comment, and have modified the text throughout to enhance clarity. We will note that we mention "liquid" in several contexts: 1) in the context of the general thermodynamic theory of the cenotectic, wherein we are referring to an arbitrary liquid phase; 2) in the context of our experiments, where we are referring to an aqueous liquid phase; and 3) in the context of planetary science/the oceans of icy moons, where we are referring to an aqueous liquid phase. We have clarified this throughout, and made sure that, in the select instances wherein we discuss pure water as the liquid phase, that too is clear.

With regard to clarifying that the solutes considered in our experiments here are exclusively salts, we have updated "solute-bearing solid phase" to "salt-bearing solid phase" as appropriate, and otherwise worked to clarify this topic throughout.

(DONE)

2. The authors have attempted to improve the clarity of the manuscript by including the methodology. This is however given at the end of the text with no reference to it in the results section that starts immediately after the introduction. One of the main contributions of the manuscript is the establishment of a new method. Therefore, I strongly feel that the methodology section should be before the results; the reader needs to know how the experiments were conducted before the outcomes of the experiments are discussed.

We thank the reviewer for this comment. In our presentation of the methodology, we have sought to follow the formatting guidelines of *Nature Communications* (<https://www.nature.com/ncomms/submit/article>), which recommend that the text be organized as Introduction → Results → Discussion → Methods. The Methods section is also considered separate from the Main Text, subject to a separate word limit, etc. The

organization of our manuscript is also consistent with other recent *Nature Communications* publications (token example: <https://www.nature.com/articles/s41467-024-50839-3>). As such, we suggest respectfully that we must keep the Methods section where it is at present. However, we have modified the initial paragraphs following the Introduction both to introduce the methodology at a high level and to emphasize to the reader that the complete methodology is available in the Methods section. We hope this strikes an appropriate balance, and thank the reviewer for their comment here.

3. The cooling and subsequent heating profiles have been given in the experimental section. How do the authors know that they are at equilibrium, especially at the start of the experiment. If the mixture is cooled down for only 45 min, is this enough to get to equilibrium? Personal experience suggests that equilibrium, especially those including a phase change where there is no mixing can take very long to attain, hours, sometimes even days, not just minutes. Figures S3 and S5 do not start at the same pressure when heated; is this not an indication that the experiments were not at equilibrium? I am not convinced regarding the results for the salts in Figure 2(c).

We very much appreciate the reviewer's comment here, which identified some typographical errors in our main-text Methods section that were inconsistent with the data and description we provide in Supplementary Note.

Critically, the solutions are cooled for much longer than 45 min. This was shown/described in Supplementary Note 2, but our according description in the main-text Methods section contained some confusing language that obscured this fact, which we have now corrected. We wholeheartedly agree with the reviewer that 45 minutes on cooling would high-certainly be insufficient initial equilibration time, especially for so large a system (~5mL). **As shown in Fig S2**, upon cooling, we monitor the pressure signal to ensure that the ice II/III transition occurs (as indicated by a precipitous collapse in the pressure), then leave the system to equilibrate for several additional hours before initiating the warming protocol. We have modified the Methods section text accordingly—most importantly, the description of the freezing and melting process has been modified to include the following, consistent with our Supplementary notes and figures:

“Freezing process

Two high-accuracy recirculating cooling baths were employed for isochoric testing, a PolyScience AP15R-40 with a minimum working temperature of 243.15K and a Fluke Calibration 7380 with a minimum working temperature of 193.15K. The latter was utilized for solutions with phase transitions lower than 243.15K (including all those exhibiting intermediate high-pressure hydrate phases), and the former for all others. For each

solution, three replicate samples in identical chambers were studied, and the chambers are submerged completely within the cooling bath fluid to ensure uniformity of temperature. More details on the cooling methodology and equipment are available in Supplementary Note 1, and a sample cooling profile is shown in Supplementary Note 2.

For solutions without intermediate high-pressure hydrate phases (i.e. those with monotonic P-T trajectories), the chambers are plunged into a bath pre-cooled to at least 30K beneath the 0.1 MPa eutectic temperature of the solution, monitored until the ice III/II transition is observed (indicated upon cooling by a precipitous collapse in pressure to approximately 210 MPa; see Fig. S2), and then allowed to further equilibrate until the pressure stabilizes to <0.1 MPa/min variation (typically 1-2 hours).

For solutions found to host intermediate high-pressure hydrate phases (NaCl, NaHCO₃, and MgCl₂, i.e. those with non-monotonic P-T trajectories), contrary to their less phase-diverse counterparts, preliminary testing showed broad varieties in behavior sample-to-sample, leading us to believe that metastable configurations were being produced and that a more gradual cooling approach may be necessary. As such, for the protocol identified as “Slow Cooling” in Figure 2c., the chambers were pre-cooled only to 273.15K, and the bath temperature was then decreased by 1K every 10 minutes. The protocol identified as “Fast Cooling” follows the approach described above for solutions without high-pressure intermediates, wherein the chamber is submerged directly into liquid pre-cooled to the minimum temperature of the run, with the exception that pressure equilibration after the ice III/II transition typically required 6-8 hours. In order to demonstrate the hallmark dependence of metastable configurations on thermal history, a combined slow-fast cooling approach was also tested, which incorporated a pseudo-annealing logic. In this process, the temperature was incrementally lowered at a rate of 1K per 10 minutes until the intermediate hydrate was detected, then raised 5K above the observed transition temperature, and cooled once more at a rate of 0.5K/min to the cenotectic point.

See Supplementary Note 2 for example time-series data on the freezing process, and Supplementary Note 3 for additional data on solutions with intermediate hydrates.

Melting process

For all solutions, after completion of the cooling process, the system is heated in discrete increments of 0.5K, with time increments chosen based on the heat transfer characteristics of each bath to produce steady steady-state P-T data by the end of each step (at which point the data is sampled), as indicated by a <0.1 MPa/min evolution of the pressure. Additional equipment details are available in Supplementary Notes 1 and 2, and an example cooling-warming profile is shown in Figure S2. The steady pressure and

temperature are recorded at the end of each step, and this pressure-temperature data provides the P-T curves shown in Figure 2.”

With regard to the results shown in Fig. 2(c), and supplementary Figs. S3 and S5:

These figures all show examples of the “non-monotonic” T-P profiles we observed, i.e. results that suggest the formation of intermediate high-pressure hydrate phases that may be metastable and confound the thermodynamic path towards the cenotectic. As described in the text, we are confident that we have *not* observed the cenotectic in these systems, because of the **cooling-rate dependent formation of metastable assemblages**. The different pressures observed at the initiation of warming and the differences in T-P profile between like samples are both consequences of this metastability, as is the acute effect of cooling protocol. These observations both convince us that these non-monotonic profiles have likely *not* reached global equilibrium, and that, by virtue of contrast, the other monotonic profiles *have* reached equilibrium. For more discussion of these intermediate hydrate phases and metastable phenomena, please see the section “**Intermediate salt-based phase transitions along the eutectic curve**”. We have also added a statement to the end of this section highlighting additional kinetic aspects of the systems that may be interrogated to explore their potential metastability for deeply.

We sincerely thank the reviewer again for their comments here, which, in both this review and the previous, have no doubt strengthened the manuscript. We hope that these clarifications of the methodology and interpretation satisfy their remaining concerns.

Some minor comments

4. Please add page numbers and give figures and tables as soon as possible after first mention. *We have done so—thank you for noting.*

5. The figure captions are very long. Be careful of including descriptions and interpretations in the captions. *Thanks again for the flag—we have cut ~4 lines from the Figure 2 caption (removing interpretational elements), and tightened up each of the others.*

RESPONSE TO REVIEWERS

We would like to sincerely thank Dr. Driesner for examining our manuscript once more, and especially within the context of evaluating our response to the concerns of Reviewer #5, and we're pleased to hear that he has found those concerns sufficiently addressed. Regarding his new insights into Figure 1, we reproduce his comments below in black and respond to them in blue.

REVIEWER 3

I (Reviewer 3) was asked for a final check if another reviewer's (5, who was not available) points were properly addressed in the revision. I am happy to say that a careful inspection of the revision lets me conclude that this is the case.

Pleased to hear it, thank you for taking the time to do that.

However, when re-reading the manuscript with all the new clarification I felt that a few things remained unclear to me. If you combine Fig. 2 with Fig. 1 you get the following: [image not shown]

Clearly, this looks as if something is wrong with the interpretation as the melting paths (arrows) start way off any possible cenotectic of the kind drawn. It would also not explain the near-horizontal initial pressure curves as those should be along the variable-pressure Ice Ih/II boundary.

After thinking and trying out possible explanations (2 – 3 hours, something you don't want a reader to spend to understand your results) I finally mean to have understood that Fig. 1 is actually misleading. In fact, the "solubility surfaces" of your systems are located much higher up, at low to very low solute mole fractions (please convert your wt% to mole fraction to see yourself).

So, I think what you want to draw is rather something like my second figure (of course nicer and more accurate than my quick sketch). Also, the cenotectic pressures now make sense as they are related to Ih/III without any Ih/II influence. When you convert to mole fraction (perhaps taking into account ion dissociation for the salts vs. urea) and consider the 0 bar ice melting curve as a generic, colligative property-based curve, then this also nicely explains why you have two groups of curves in Fig. 2a, and perhaps also the "22K rule of thumb".

I would strongly recommend to think this through and – if you think I am right – consider revising the manuscript accordingly, including possible newly gained insights. After such a revision the manuscript is very likely ready for publication.

We thank the reviewer tremendously for this analysis and insight, and wholeheartedly agree that the figure was unnecessarily confusing-- because indeed none of the systems shown in Fig. 2a/b **HAVE** their cenotectic in equilibrium with ice III! We have modified the figure to take the valuable noted insights into account. We have shifted the representation to a higher range where only ice III is present at the cenotectic, consistent with the data shown in Fig. 2a/b, and we have revised the figure caption to enhance clarity (see the new figure and caption at end of this document).

We note further that this figure is only a schematic aimed at illustrating the principle of the cenotectic concept, not an empirically accurate representation of the liquidus and solubility curves of any particular system investigated herein-- hence the use of the a.u. in the composition space. To clarify this further, and remove any potential misconceptions related to units or scale of concentration (mass % vs. mol %), we have also removed "1" at the right-hand side of the concentration axis.

We have also combed the manuscript to ensure that this edited figure is in no way inconsistent with other written aspects of our interpretation, and have found that it is not. Throughout the results section, we discuss the cenotectic point for the systems at hand as either occurring at the liquid / ice Ih / ice III intersection (as is certainly the case in the uncomplicated binaries shown in Fig 2a/b), or possibly at the liquid / ice Ih / ice II intersection for the systems displaying intermediate hydrate phases in Figure 2.c. As such, the Figure 1 now reflects the case for which we have the most (and most definitive) data, and should accurately prepare the reader for the data presented in Figure 2.

We hope that these edits makes the figure (and manuscript!) considerably less confusing, and we would like to sincerely thank Dr. Driesner again for this suggestion. In light of his above-and-beyond reviewer input here, we would also like to credit him in the acknowledgment of this manuscript (with his and the editor's permission).

As a final remark (also to the editor, given the response on paper format in the rebuttal): you may now see why it is so important to really clearly document your experimental approach well and right from the beginning.

Sincerely, Thomas Driesner

We thank the reviewer for this comment as well, and certainly agree that better organization upon initial submission could no doubt have saved us (all) significant time and effort—we will bear it mind going forward! Thanks again for a very valuable review.

Revised Figure 1 and caption:

Figure 1: Conceptual pressure-temperature-concentration phase diagram for a binary solution of water and a generic salt-like solute with only one solute-bearing solid phase, with the eutectic at ambient pressure marked (**e**), the pressure-dependent eutectic curve marked by the thick magenta line, and the cenotectic point (κ) marked as a star. m_e marks the brine concentration at the 0.1 MPa eutectic, and m_κ marks the brine concentration at the cenotectic point. P_κ, T_κ mark the pressure and temperature of the cenotectic point, which we measure in this work. Isochoric freezing/melting of brine samples of concentration m_e generally follows the pressure-temperature trajectory marked by the magenta eutectic curve between points (**e**) and (κ), and the cenotectic point can be identified by the change in direction of the P-T slope of this curve.

RESPONSE TO REVIEWER

We thank Dr. Driesner once more for his sustained effort in reviewing and improving our manuscript. We have addressed his remaining comments in the final version of our manuscript. For the editor's ease, we reproduce Dr. Driesner's comments below (blue) and the according changes we've made to the manuscript (black). These changes are also highlighted in the "marked-up" version of the revised manuscript file. We thank the editor and reviewer again for their terrific effort here.

Reviewer #3 (Remarks to the Author):

Thanks for the positive feedback on my previous comments, and I'd be happy to be acknowledged.

Obviously the revised figure fits the data and text doesn't contradict the new Figures, but: I wonder if you don't give away an opportunity to show the (claimed) generality and to make even some predictions. Sorry for bringing that up as the quick response seemed to indicate some pressure to get the paper out. Yet, I think for a "high impact" journal also the content should make as much out of the content as possible with the insights obtained during the study. And, I like the results and would find it a pity if you let the chance pass to make this more "seminal".

So, my previous suggestion with three surfaces on Fig. 1 showed that the experimental curves could align differently (quite significantly, actually, your Figure 2) with the ice diagram with the mole fraction of solute particles. Assuming full dissociation of electrolytes and none of urea, these are MgSO_4 0.03, Na_2SO_4 0.016, Na_2CO 30.03 and KCl 0.11, urea 0.127, if I didn't miscalculate that. One sees easily, that these numbers align nicely with the positions on Fig 2a and I think there is quite some benefit from adding this insight to Fig 1.

One can now speculatively predict that there may be cases (with high eutectic molar fractions of solute particles) for which one would predict a cenotectic with ice II stability (you original Fig. 1) and according possible variations in cenotectic pressure etc.

I attach a slightly changed suggestion for what could be modified in Fig 1 to highlight this (and it needs a disclaimer that it is schematic; yet, all salts except MgCl_2 seem to follow a +/- similar eutectic curve at 0.1 MPa if plotted in mole fraction space and assuming full dissociation, so it seems very acceptable to stick to the suggested design). Please consider this seriously as a starting point for a revision of Fig. 1 given the points above (and a few minor explanations added to the text and maybe in the results).

Of course, it is your paper; if you feel that the above thoughts are misleading or likely/possibly "too wrong" to be considered, then please reject the suggestion but with a good explanation why.

Response:

We agree completely with Dr. Driesner that the "expanded" version of Figure 1 he suggests will deepen the impact and clarity of the work. As such, we have elected to capture the best of all worlds in the final revision and include **both** our Figure 1 from the previous revision (which the Reviewer notes is now consistent with our data in Figure 2) **and** the "expanded" version suggested by Dr. Driesner, which we include as Figure 3 in the revised text. That figure and the according descriptive text added is reproduced below:

Figure 3: Conceptual pressure-temperature-concentration phase diagram depicting binary solutions of water and generic salt-like solutes of varying solubilities. A generic low solubility surface is shown in red, a generic medium solubility surface is shown in yellow, and a generic high solubility surface is shown in green. The 0.1 MPa eutectic points corresponding to each surface are marked by (e_1 , e_2 , e_3), and the centenectics are marked by (κ_1 , κ_2 , κ_3), where increasing subscript indices correspond to increasing solubility. All other markings retain the meanings indicated in Figure 1.

For clarity, in Figure 3, we have extended the conceptual phase diagram presented in Figure 1 to include prototypical examples of solute solubility surfaces that may correspond to these differing centenectics, organizing them by molar solubility. Examples of low solubility (e.g. aqueous Na_2SO_4 or MgSO_4) and medium solubility (e.g. aqueous KCl or Urea) centenectics are represented within the experimental data in Fig. 2a/b, and marked conceptually in Fig. 3 by κ_1 and κ_2 . Examples of high solubility centenectics (occurring at the ice Ih-ice II-liquid triple point and marked by κ_3) have not been experimentally confirmed in this work, but we suggest that aqueous perchlorates may be good candidates, as they often possess very low 0.1 MPa eutectic temperatures (e.g. 236 K for NaClO_4 and 206 K for MgClO_4). For aqueous NaCl and MgCl_2 , while we observe signs of new hydrate species at high pressures that may increase the temperature of the centenectic, it appears likely that they will yet fall beneath 238K (along the Ih-ice II-liquid triple point), considering the trends of their eutectic curves (Fig.2c).

We agree with Dr. Driesner that this figure best conveys the broadness of the possibility space, and will help the reader to understand the wide array of centenectic possibilities presented by different aqueous systems.

Review of the revised manuscript by Zarriz et al. “On the equilibrium limit of liquid stability in pressurized aqueous systems”

I (Reviewer 3) was asked for a final check if another reviewer’s (5, who was not available) points were properly addressed in the revision. I am happy to say that a careful inspection of the revision lets me conclude that this is the case.

However, when re-reading the manuscript with all the new clarification I felt that a few things remained unclear to me. If you combine Fig. 2 with Fig. 1 you get the following:

Clearly, this looks as if something is wrong with the interpretation as the melting paths (arrows) start way off any possible cenotectic of the kind drawn. It would also not explain the near-horizontal initial pressure curves as those should be along the variable-pressure Ice Ih/II boundary.

After thinking and trying out possible explanations (2 – 3 hours, something you don’t want a reader to spend to understand your results) I finally mean to have understood that Fig. 1 is actually misleading. In fact, the “solubility surfaces” of your systems are located much higher up, at low to very low solute mole fractions (please convert your wt% to mole fraction to see yourself).

So, I think what you want to draw is rather something like my second figure (of course nicer and more accurate than my quick sketch). Also, the cenotectic pressures now make sense as they are related to Ih/III without any Ih/II influence. When you convert to mole fraction (perhaps taking into account ion dissociation for the salts vs. urea) and consider the 0 bar ice melting curve as a generic, colligative property-

based curve, then this also nicely explains why you have two groups of curves in Fig. 2a, and perhaps also the “22K rule of thumb”.

I would strongly recommend to think this through and – if you think I am right – consider revising the manuscript accordingly, including possible newly gained insights. After such a revision the manuscript is very likely ready for publication.

As a final remark (also to the editor, given the response on paper format in the rebuttal): you may now see why it is so important to really clearly document your experimental approach well and right from the beginning.

Sincerely, Thomas Driesner